# Simultaneous measurement of greenhouse gases (CH$_4$, CO$_2$ and N$_2$O) using a simplified gas chromatography system

Michał Bucha[1], Dominika Lewicka-Szczebak[1], Piotr Wójtowicz[2,3]

[1] Institute of Geological Sciences, University of Wrocław, pl. M. Borna 9, 50-204 Wrocław, Poland
[2] Department of Environmental Engineering, West Pomeranian University of Technology, Piastów 45, 70-311 Szczecin, Poland
[3] Shim-Pol A.M. Borzymowski, Official Shimadzu Distributor in Poland, ul. Kochanowskiego 49A, 01-864 Warszawa, Poland

*Correspondence to*: Michał Bucha (michal.bucha@uwr.edu.pl)

**Abstract.** This article presents a simple method for determining greenhouse gases (CH$_4$, CO$_2$, N$_2$O) using an alternative new set-up of the chromatographic system. The novelty of the presented method is the application of a Carboxen 1010 PLOT capillary column for separation of trace gases – CH$_4$, CO$_2$ and N$_2$O – from air samples and their detection using a barrier discharge ionisation detector (BID). Simultaneously, a parallel molecular sieve column RT-Msieve 5A connected to a thermal conductivity detector (TCD) allowed the determination of CH$_4$, N$_2$ and O$_2$ concentrations from 0.2 to 100 %. The system was equipped with an autosampler transferring the samples without air contamination thanks to a vacuum pump-inert gas flushing option. Method validation was performed using commercial gas standards and undertaking a comparison measurement with a reference method: optical methods applying Picarro isotope and concentration instruments with cavity ring-down spectroscopy for CO$_2$, CH$_4$ and N$_2$O. A three-day continuous measurement series of GHG concentrations in ambient air and tests of typical vial sample measurements with increased GHG concentrations were performed.

The advantage of this method is that the system is easy to set up and allows for simultaneous detection and analysis of the main GHGs using one GC column and one detector, thereby omitting the need for an electron capture detector (ECD) containing radiogenic components for N$_2$O analysis and a flame ionisation detector (FID) with a methaniser for low-concentration CO$_2$ samples. The simplification of the system reduces analytical costs, facilitates instrument maintenance and improves measurement robustness.

## 1 Introduction

The reduction of greenhouse gas (GHG) emissions caused by human activity presents a major challenge that needs to be addressed in order to limit the effects of global warming. The main GHGs responsible for global warming are CO$_2$, CH$_4$ and N$_2$O (Lamb et al., 2021). Besides natural sources (e.g. volcanic activity, peat bogs, paddy soils, fresh and saltwater sediments etc.), human activity also contributes to increasing GHG emissions by having an impact on global carbon and nitrogen cycling.

Therefore, precise measurement of GHG concentrations from natural sources and the environment is crucial in order to quantify and estimate the contribution of different anthropogenic sources to worldwide emissions. The development of analytical equipment in recent years has allowed the application of user-friendly methods to determine trace gases and monitor slight changes in their concentrations precisely, even at the lowest levels expressed in units ppm and/or ppb (Zaman et al., 2021). In the very near future, it can be expected that analytical devices and their measurement precision will be enhanced further. Therefore, in order to maintain reliable continuity of measurement data on GHG concentrations in the atmosphere and in other elements of the Earth's ecosystem, measurements should be performed with the utmost care using the most modern techniques and devices available. It is also important to maintain easy access to simple and relatively cheap analytical devices and their ease of use in order to obtain more statistical data. Often the analytical devices providing very precise measurements have limitations in the analytical range and do not allow for observations of GHGs in a wide range of concentrations observed in nature. Therefore, the development of a new analytical method that is characterized by relatively high sensitivity in the range from the lowest to the highest concentrations is a desirable feature, but quite difficult to achieve in a single device.

Several methods are available for monitoring GHG based on optical techniques such as non-dispersive infrared spectroscopy (NDIR), Fourier-transform infrared spectroscopy (FTIR), photoacoustic spectroscopy (PAS), tunable laser absorption spectroscopy (TLAS), cavity ring-down spectroscopy (CRDS), and off-axis integrated cavity-output spectroscopy (OA-ICOS) (Zaman et al., 2021). Some of these laser instruments allow for simultaneous analyses of $CH_4$, $CO_2$, $N_2O$ and $NH_3$ using the laser absorption spectroscopy method (e.g. Picarro G2509 Gas Concentration Analyzer). Although these devices guarantee the stability of continuous measurements, their measurement range is much lower than chromatographic systems equipped with typical detectors. For example, the Picarro G2509 operation range for $CO_2$ is 380-6000 ppm, for $CH_4$ 1-800 ppm and for $N_2O$ 0.3-200 ppm. Other versions of Picarro analysers have been developed to measure single GHG concentrations, e.g. of $CH_4$, $CO_2$ or $N_2O$, in combination with analyses of stable isotope composition of carbon or nitrogen from atmospheric air or headspace samples (SSIM module). These methods are recommended only for the measurement of a single gas compound at very specific concentrations. Thus, the most reliable methods for GHGs measurements in a very wide range of concentrations are chromatographic methods (Ekeberg et al., 2004). Another important limitation of the laser-based system is the sample matrix, which should be stable and most similar to standard ambient air composition. Hence, these methods are not well suited for untypical gas samples, like e.g. mine gases, or samples originating from laboratory experiments, e.g. with He atmosphere.

Gas chromatography with automated sampling injections is a very common, flexible and user friendly technique. The most common GHG measurement systems have been developed with: a thermal conductivity detector (TCD) (measurement of $CH_4$ and $CO_2$), flame ionisation detector - FID (measurement of $CH_4$ and $CO_2$ using a methaniser), electron capture detector - ECD (for $N_2O$ measurement) (Hedley et al., 2006; Loftfield et al., 1997; Wang and Wang, 2003). The GC systems can be dedicated for specific gases at ambient concentrations with precision similar or even better to those achieved by optical techniques. Van der Laan et al. (2009) developed the GC system which allows for simultaneous measurement of $CH_4$, $CO_2$, $N_2O$, CO and $SF_6$ using one gas chromatograph and single injection, that allows the measurement of GHGs from ambient air at remote stations. However, the system is characterised by a quite complex set-up with multiple gas valves, columns, methaniser, ECD and FID.

The most popular analytical technique for determining $N_2O$ concentration is gas chromatography equipped with ECD using Porapak Q or Hayesep Q columns (Rapson and Dacres, 2014). However, the use of an ECD is associated with additional difficulties. The main disadvantage of the ECD is its poor stability over a long period of time. During ongoing analyses, the cell interior may become contaminated and natural wear may occur. This can result in an increasing response to the tested concentrations. Consequently, within a week a significant increase in the measured area may be observed for the same analysed concentrations. This drift can be compensated for by the addition of an internal standard. Moreover, due to the presence of radioactive material in the ECD, special safety requirements have to been taken into account. According to current regulations, the purchase of a new unit, its possession and the disposal of a used detector cell involve a number of formal requirements.

In the case of the dielectric barrier discharge ionisation detector (BID), there are no such limitations and restrictions. The only requirement is to ensure a supply of helium of appropriate purity (99.9999 %). The detector is incredibly stable and maintenance-free for a very long period of time. Application of the BID for $N_2O$ measurements has the advantages of avoiding radiogenic compounds present in the ECD and reducing the number of gases required. Combination of ECD+FID requires installation of minimum 3 gas tanks (carrier gas He, Ar, or $N_2$; synthetic air; $H_2$ or $H_2$ generator; make-up gas for ECD $N_2$ of 6.0 purity, min. 99.9999%), whereas for BID only He tank is required.

Separation of $CH_4$, $CO_2$, and $N_2O$ from one sample can be done using, for example, a system of two columns with 10-port valves (Scion Instruments, 2023) or a single column e.g. Micropacked ST Shin Carbon or RT Q-Bond column (Shimbo and Uchiyama, 2022). Methods using a single-column Micropacked ST Shin Carbon or RT Q-Bond are typically applied by Shimadzu using a GC Nexis 2030 gas chromatograph equipped with a dielectric barrier discharge ionisation detector (BID) dedicated to trace compounds (Shimbo and Uchiyama, 2022). This set-up using single column and single BID detector is commonly used for determination of $CH_4$ and $CO_2$ at very low atmospheric concentrations (Gruca-Rokosz et al., 2020). Unfortunately, the retention times for $CO_2$ and $N_2O$ are often insufficient for correct measurement, especially by high $CO_2$ concentrations, when $CO_2$ tailing can even cover the $N_2O$ peak. This separation can be enhanced by application of cryogenic methods for decreasing oven temperature. However, these methods are time-consuming and expensive. The present study tested an alternative solution that involved the application of Carboxen 1010 PLOT for $CH_4$, $CO_2$ and $N_2O$ separation.

A simple chromatographic system is presented here for quick and accurate analysis of GHG using TCD and BID of the GC Nexis 2030 gas chromatograph combined with an AS-210 Greenhouse Gas Autosampler (SRI Instruments Europe GmbH, Bad Honnef, Germany) at wide range of concentrations from ambient observed in the Earth's atmosphere to the higher fluxes observed typically for the measured emission sources.

The GC separation columns used in this study were performed with a porous layer open tubular column (Carboxen 1010 PLOT) and a molecular sieve column (RT-Msieve 5A), which assured the full separation of the analysed gases. The results of the experimental data were compared with the concentrations obtained for $CH_4$, $CO_2$ and $N_2O$ using the CRDS technique by Picarro analysers (G2201-i for $CO_2$ and $CH_4$; G5131-i for $N_2O$).

## 2 Materials and methods

### 2.1 Gas chromatography system

This chromatographic system was built based on the Shimadzu GC Nexis 2030 equipped with two parallel detectors: BID and TCD (Fig. 1). The carrier gas was controlled by an advanced flow controller (AFC) connected to a split/splitless injector. Between the AFC and the injector, a two-position six-port valve with a 1 mL (or 2 mL) sample loop was placed on the carrier line.

The gas chromatograph oven was equipped with an additional cryogenic option (CRG) where liquid nitrogen ($LN_2$) was used as a cooling agent, which allowed for separation at below-ambient temperatures. The samples from AS-210 Greenhouse Gas Autosampler (SRI Instruments GmbH) were transferred to the sampling valve through a stainless steel transfer line continuously warmed to 110°C with heating tape to prevent moisture contamination. The presence of moisture in the samples results in loss of sorption capacity of carbon molecular sieves which are used commonly for separation of gases (Fastyn et al., 2003). The injection was performed by valve rotation. The sample was transferred from the loop (1 mL or 2 mL) through the injector at a total flow of 10 mL/min, and was then split 1:7 just before the column inlet. This was sufficient to achieve a good peak shape with sufficient area.

Additionally, using a T-joint, the injection sample was then divided between two porous layer open tubular capillary columns filled molecular sieve 5A (RT-Msieve 5A 30 m x 0.32 mm x 30 μm; Restek, USA, Cat. No. #19722) and fused silica (Carboxen 1010 PLOT 30 m x 0.53 mm x 30 μm; Supelco, USA, Cat. No. #25467). The dimensions of the columns were selected to achieve a splitting ratio of 1:5, directing most of the sample to the Carboxen 1010 PLOT and BID. Corresponding calculations were performed in Shimadzu AFT (Advanced Flow Technology) software (Fig. 2).

Extremely low baseline noise (signal/noise ratio (S/N) always above 10) was achieved by a combination of two factors: a high purity carrier gas helium grade 5.0 connected to the Valco Helium Purifier HP2 (VICI Valco Instruments Co. Inc.) and particle traps (2.5 m x OD 0.32 mm) mounted on the columns' outlets. The presence of the traps protected the detectors from particles dislodging from the porous layer open tubular (PLOT) capillary column, which can cause spikes.

Both the detectors used are concentration dependent; therefore to obtain the highest sensitivity on the BID channel, the discharge gas flow rate was decreased from a default of 50 mL/min to 30 mL/min, which is the lowest possible flow (Barrier Discharge Ionization Detector for GC-2010 Plus BID-2010 Plus Instruction Manual. Shimadzu Corporation 2013). Below this value the plasma flame is not stable and tends to flicker or extinguish. Detection levels did not need to taken into consideration with the TCD

The linearity of the both detectors response was controlled and maintained during all the measurements with minimal $R^2=0.99$ applying at least two standard gases and 0 point.

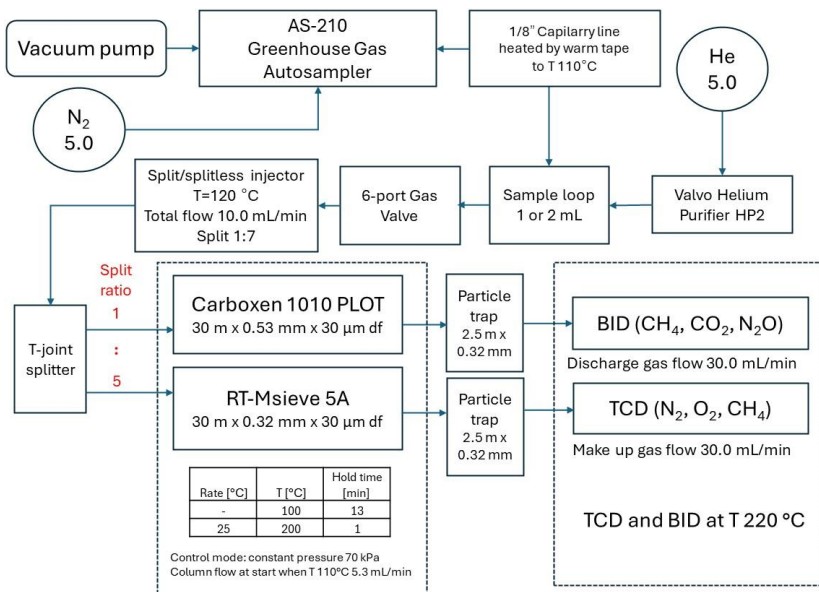

**Figure 1: Configuration of GC system for measurement of CH₄, CO₂ and N₂O**

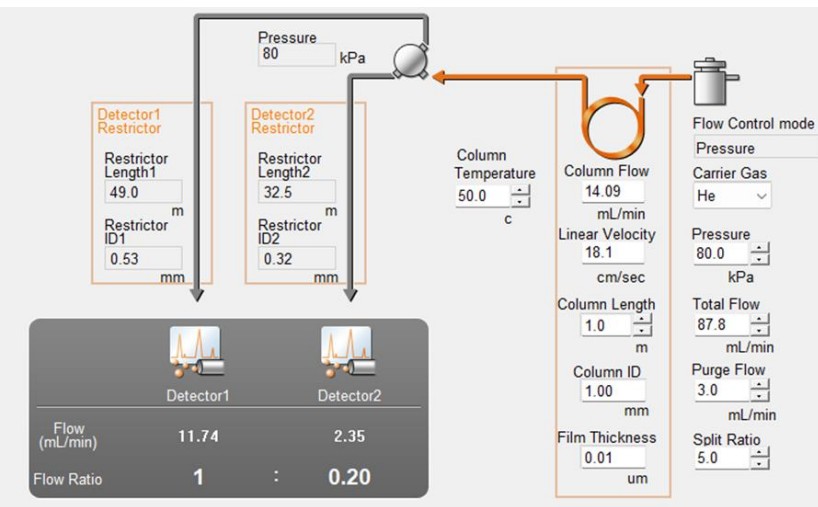

**Figure 2: Detailed flow parameters in GC system configuration**

## 2.2 Parameters of the separation and detection methods

The temperature programme for gas chromatography analyses started at 100 °C for 13 minutes, and later increased to 200 °C at a rate of 25 °C/min with the oven set at 200 °C for one minute. The temperature of the split/splitless injector was 120 °C. The TCD and BID were at an equal temperature of 220 °C. The carrier gas pressure was 70 kPa and the column flow was 5.3 mL/min. Linear velocity was 41.2 cm/s and purge flow was 1 mL/min. The total flow for split ratios 1, 2, 3, 4 and 5 was 11.5, 16.8, 22.0, 27.3 and 32.5 mL/min, respectively.

### 2.3 Standard gas mixtures

Standard gas mixtures used for testing and final determination of the measurements precision were atmospheric air from Wrocław (Poland) (analyses of $N_2$, $O_2$, $CH_4$, $CO_2$ and $N_2O$ at ambient atmospheric concentrations) and a special gas mixture from Messer ($CH_4$ 10 ppm, $CO_2$, 1000 ppm, $N_2O$ 50 ppm, diluted in pure $N_2$). The in-house standard of compressed air from Wrocław (Poland), which contained natural moisture (water vapour), was stored in the 10 litres gas cylinder. It was prepared by oil-free compressor for diving cylinders. The second standard was ordered in Messer Polska Sp. z o.o. and is the commercial product prepared in Switzerland according to the norm ISO6141:2015. This standard was prepared in pure $N_2$, without moisture, in volume of 8 litres and contains F10 filter, which protects outer valve from the possible water vapour or solid particles. The standards were directly connected by 1/8" capillary to the AS-210 Greenhouse Gas Autosampler. The sample loops used for tests of standard gases were 1 mL and 2 mL. The atmospheric air was tested for splits 1, 2, 3, 4 and 5. The special gas mixture from Messer was tested for splits 3, 4 and 5.

## 3 Results and Discussion

### 3.1 Basic testing of the SH-Q-BOND and Carboxen 1010 PLOT columns and BID detector

The chromatographic system for GHG analyses using a single BID was initially tested for application of the SH-Q-BOND column (30 m x 0.53 mm ID x 20 um df; Cat. No. #221-75765-30) from Shimadzu, which allows for separation of $CH_4$, $CO_2$ and $N_2O$, and is resistant to water vapour contamination. The scheme showing GC configuration for testing of SH-Q-BOND and RT-Msieve 5A is presented in Appendix A (Fig. A1). In this configuration most of the parameters were exactly the same (length of the column, diameter, film thickncess, flow parameters, split ratio) as in configuration using Carboxen 1010 PLOT column. The only differences were usage of SH-Q-BOND column for separation of $CH_4$, $CO_2$, $N_2O$ and different column oven temperature programme.

Separation of these gases was tested at different low temperatures of the column oven (30 °C, 35 °C and 40 °C). The lowest temperature (30 °C) was difficult to achieve quickly by the oven without using a cryogenic trap. At the laboratory's normal temperature (22 °C), it was possible to decrease the oven temperature rapidly to 35 °C and 40 °C, but unfortunately both temperatures were insufficient to separate $CO_2$ from $N_2O$ at a retention time interval longer than 30 seconds, which appeared very problematic while analysing real samples of elevated $CO_2$ concentration. When the $CO_2$ concentration was high (e.g. 700 ppm), the tail of the $CO_2$ peak partially covered the $N_2O$ peak, as shown below in Figures 3A and 3B, and ultimately the $N_2O$ peak area was understated. Moreover, this set-up did not allow detection of $CH_4$ in atmospheric concentrations.

**3A**

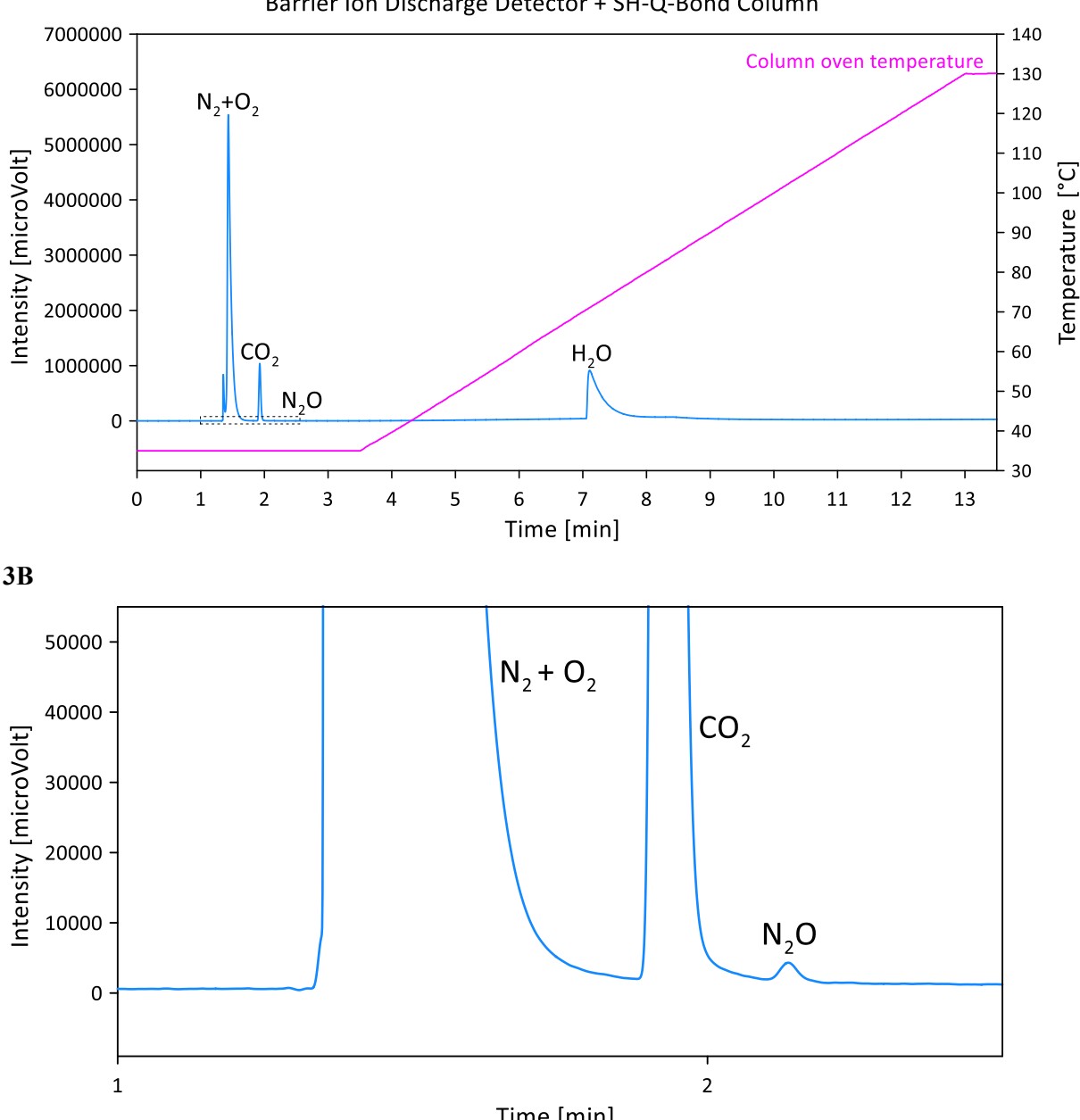

Figure 3: A - Chromatograms of ambient air gases separated using the SH-Q-BOND column and detected using BID;

**3B**

**Figure 3: A - Chromatograms of ambient air gases separated using the SH-Q-BOND column and detected using BID; B – Zoomed chromatogram from Figure 3A**

Therefore, after basic tests of the SH-Q-BOND column at different temperatures, it was decided to check the retention times

of individual $CH_4$, $CO_2$ and $N_2O$ gases on the Carboxen 1010 PLOT column. The Carboxen 1010 PLOT column offered very
good separation of $CO_2$ from $N_2O$, even at a very high concentration ($CO_2$ 1000 ppm), as shown in Figure 4 (peaks of $CH_4$ and
$N_2O$ are visible only when zoomed). Moreover, the ambient $CH_4$ was very well separated from the $N_2+O_2$ peak (Fig. 4). The
longer programme of separation guaranteed the ideal separation of $CH_4$, $CO_2$ and $N_2O$ (more than two minutes between each
gas). The time of one single analysis is 18 minutes, but this is necessary because a low flow of the carrier gas is recommended

for the Carboxen column by its manufacturer. A carrier gas flow that is too high (e.g. above 20 mL/min) causes faster
destruction of the column and contamination of the particle trap, and subsequently of the detector, with fragments of the column
filling. However, the disadvantage of the application of the Carboxen column is also its low resistance to water vapour.
Therefore, before starting the analyses, all lines of the AS-210 Greenhouse Gas Autosampler as well as the sample loop were
carefully heated using heating tape and a gun heater to remove water from the stainless steel capillaries and metal parts of the

valves. The parallelly connected column Rt-Msieve 5A and TCD allow for determination of $N_2$ and $O_2$ (and if necessary $CH_4$)
concentrations in range from 0.2 to 100% (example showed in Fig. 5).

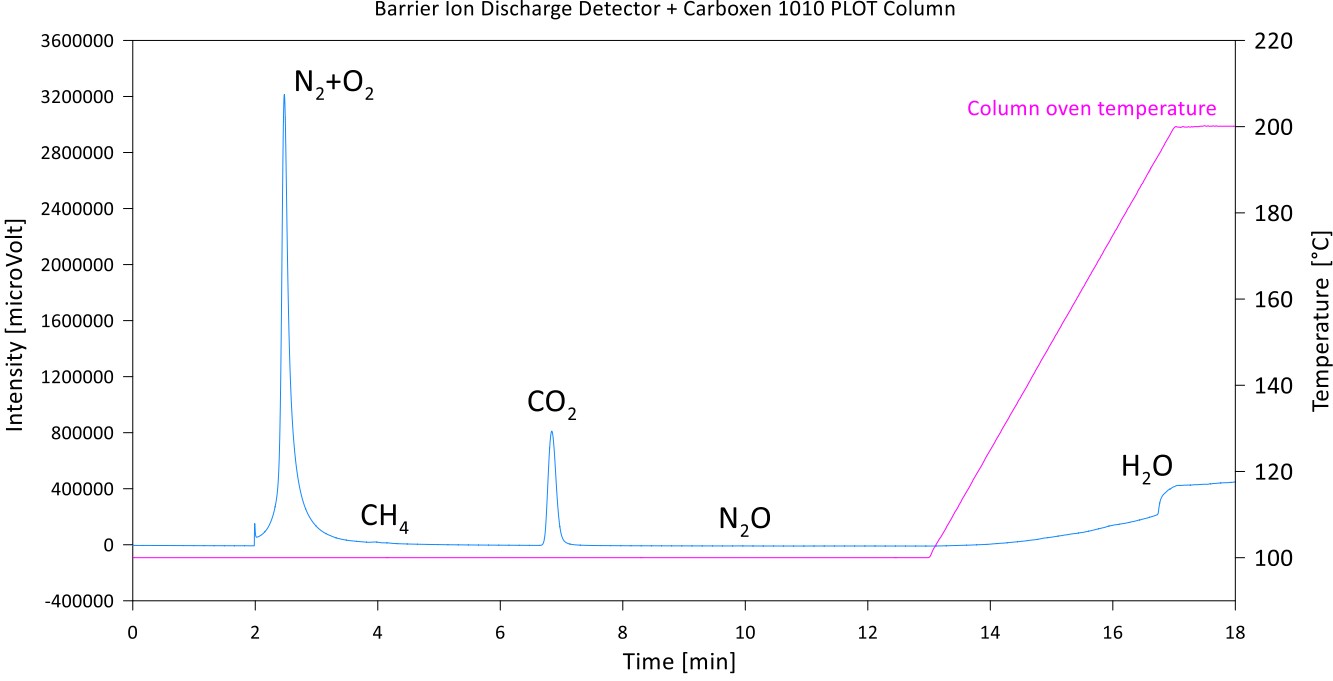

**Figure 4: Chromatogram of special gas mixture separated using the Carboxen 1010 PLOT column and detected using BID**

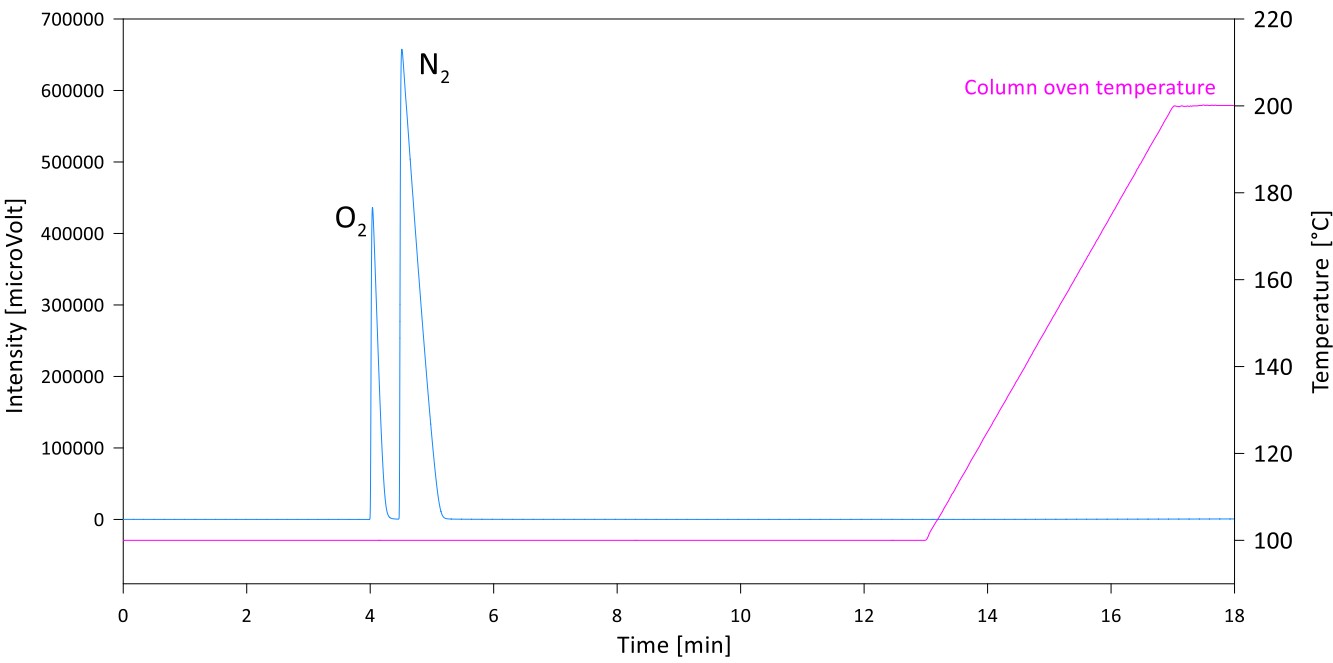

Figure 5: Chromatogram of ambient air separated using the RT-Msieve 5A column and detected by TCD

### 3.2 Compressed air standard measurements

The analyses of the compressed air standard with split ratios 1, 2, 3, 4 or 5 and using 1 and 2 mL sample loops showed a
different precision, expressed as a coefficient of variation (CV: calculated as the standard deviation divided by the mean value,
expressed in %). The CV of the $CH_4$ measurement (1.895 ppm) was in the range of 3.31 to 8.57 % for the 1 mL sample loop
and 1.51 to 2.76 % for the 2 mL sample loop. In the case of $CO_2$ (411 ppm), the CV ranged from 0.91 to 2.83 % for the 1 mL
sample loop and from 0.57 to 1.87 % for the 2 mL sample loop. The CV of the $N_2O$ measurement was lower for the 2 mL
sample loop, and ranged from 7.09 to 12.67 %, compared with the 1 mL sample loop where the CV of the $N_2O$ measurement
ranged from 9.15 to 20.19 %. Generally, it was observed that gases at low detection limits ($CH_4$, $CO_2$) were measured more
precisely using the 2 mL sample loop.

Measurement of $N_2O$ in split ratio 3 resulted in a significantly higher CV when compared with the results obtained in split
ratios 4 and 5 (Table 1). This is because the injection of a higher amount of water vapour contained in the sample partially
covered the peak area of the $N_2O$ (by increasing the baseline level), similarly to the measurement in split ratio 2 (CV 20.19
%). In this case the lower CV at split ratio 1 was only calculated for three measurements to avoid unnecessary contamination
of the column.

The gases analysed using the TCD, $O_2$ and $N_2$, were characterised by a narrow CV ranging from 0.10 to 0.39 %. The highest CV (0.39 %) was observed for the $N_2$ measurement with the sample loop 2 mL, where the peak area was very large. The results of the measurement (peak area, SD, CV) are presented in Table 1.

**Table 1: Peak area, SD (standard deviation) and CV (coefficient of variation) of standard atmospheric gas measurements at split ratios 1, 2, 3, 4, and 5 with 1 mL and 2 mL sample loops**

| Gas | Conc. | | Sample loop 1 mL | | | | | Sample loop 2 mL | | |
|---|---|---|---|---|---|---|---|---|---|---|
| | | | Split | | | | | Split | | |
| | | | 1 | 2 | 3 | 4 | 5 | 3 | 4 | 5 |
| | | Repetitions | n=3 | n=10 | n=20 | n=20 | n=20 | n=10 | n=20 | n=10 |
| $CH_4$ | 1.895 ppm | Area | 33294 | 24249 | 25271 | 20085 | 16269 | 25535 | 23758 | 23195 |
| | | SD | 2580 | 802 | 1859 | 1365 | 1394 | 386 | 381 | 640 |
| | | CV [%] | 7.75 | 3.31 | 5.95 | 6.79 | 8.57 | 1.51 | 1.59 | 2.76 |
| $CO_2$ | 411 ppm | Area | 10900323 | 8015901 | 5742835 | 4525868 | 3658827 | 10274295 | 6900992 | 6295494 |
| | | SD | 99092 | 84289 | 162281 | 58578 | 79062 | 155997 | 45587 | 117747 |
| | | CV [%] | 0.91 | 1.05 | 2.83 | 1.29 | 2.16 | 1.52 | 0.57 | 1.87 |
| $N_2O$ | 339 ppb | Area | 7801 | 4732 | 3572 | 2565 | 2080 | 5565 | 4610 | 3306 |
| | | SD | 714 | 955 | 554 | 289 | 323 | 705 | 317 | 329 |
| | | CV [%] | 9.15 | 20.19 | 15.51 | 11.26 | 15.55 | 12.67 | 7.09 | 9.95 |
| $O_2$ | 20.946% | Area | 4317080 | 3030077 | 2147582 | 1660466 | 1340040 | 4513907 | 3322830 | 2523219 |
| | | SD | 9241 | 5136 | 7177 | 4196 | 2088 | 10598 | 3298 | 7454 |
| | | CV [%] | 0.21 | 0.17 | 0.33 | 0.25 | 0.16 | 0.23 | 0.11 | 0.30 |
| $N_2$ | 78.084% | Area | 16522678 | 11587989 | 8205177 | 6343410 | 5119796 | 17338000 | 12684872 | 9709091 |
| | | SD | 42456 | 21960 | 27025 | 16376 | 8251 | 31998 | 12603 | 38248 |
| | | CV [%] | 0.26 | 0.19 | 0.33 | 0.26 | 0.16 | 0.15 | 0.13 | 0.39 |

### 3.3 Standard gas mixture measurements

The $CH_4$, $CO_2$ and $N_2O$ measurements of the special gas mixture standard at split ratios 3, 4 and 5 were characterised by a repeatable CV within a narrow range from 0.11 to 3.22 %. The CV of the $CH_4$ measurement (10 ppm) using the 1 mL sample loop ranged from 0.11 to 0.55 %, while for the 2 mL sample loop the CV was in the range of 0.34 to 1.79 %. The CV of $CO_2$

(1000 ppm) for the 1 mL sample loop was between 2.05 and 3.08 % and for the 2 mL sample loop between 1.57 and 3.22 %. The $N_2O$ measurement (50 ppm), which is a very high concentration (rare in the natural environment), was characterised by a CV in the range of between 0.18 and 0.44 % for the 1 mL sample loop and 0.85 to 2.14 % for the 2 mL sample loop. These

values clearly show that measurements of the gas mixtures with relatively high concentrations of $N_2O$ using BID were repeatable for all splits 3, 4 and 5, and were slightly better using the 1 mL sample loop. However, the application of the sample loops (1 and 2 mL) at split ratios 3, 4 and 5 guaranteed the achievement of repeatable results. Table 2 shows all the data of the standard gas mixture testing measurements.

**Table 2: Peak area, SD (standard deviation) and CV (coefficient of variation) of special gas mixture measurements at split ratios 3, 4, and 5 with sample loops 1 mL and 2 mL**

| Gas | Conc. | | Sample loop 1 mL | | | Sample loop 2 mL | | |
|---|---|---|---|---|---|---|---|---|
| | | | Split | | | Split | | |
| | | | 3 | 4 | 5 | 3 | 4 | 5 |
| | | Repetitions | n=5 | n=4 | n=3 | n=10 | n=10 | n=10 |
| CH$_4$ | 10 ppm | Area | 149922 | 118357 | 96007 | 278296 | 207227 | 161265 |
| | | SD | 831 | 133 | 318 | 1067 | 707 | 2895 |
| | | CV [%] | 0.55 | 0.11 | 0.33 | 0.38 | 0.34 | 1.79 |
| CO$_2$ | 1000 ppm | Area | 13522596 | 11079679 | 8993195 | 21485425 | 17323467 | 13525917 |
| | | SD | 331786 | 341314 | 184053 | 337763 | 331554 | 435439 |
| | | CV [%] | 2.43 | 3.08 | 2.05 | 1.57 | 1.91 | 3.22 |
| N$_2$O | 50 ppm | Area | 629251 | 497331 | 402025 | 1171624 | 867747 | 673486 |
| | | SD | 2801 | 1782 | 729 | 9975 | 11510 | 14360 |
| | | CV [%] | 0.44 | 0.36 | 0.18 | 0.85 | 1.33 | 2.14 |

### 3.4 Direct measurement of ambient laboratory air using the AS-210 Greenhouse Gas Autosampler

Another testing of the GC system was carried out with the application of splits 3, 4 and 5 and sample loop 2 ml (Table 3). The ambient air from the laboratory on one day was analysed directly from the AS-210 Greenhouse Gas Autosampler (empty plate for vials, which enabled direct sampling of the ambient air from the needle to the line connected with the GC's sample loop). The tests were performed with splits 3, 4 and 5 (splits 1 and 2 were omitted to avoid excessive introduction of air containing natural moisture into the Carboxen column).

The CV of CH$_4$ was in the range of 3.40 to 4.10 % (the highest value for split 4). The CV of CO$_2$ was in range of 1.31 to 2.29 %, whereas for N$_2$O it was between 3.11 and 4.04 %. These CV values are close to the results obtained during measurements of the compressed air standard (Section 3.3, Table 1). The difference between the two experiments is that compressed air always had the same composition and gas concentrations, whereas the CH$_4$ and CO$_2$ concentrations in ambient air could change slightly over time (daily variability).

**Table 3: Peak area, SD (standard deviation) and CV (coefficient of variation) of direct measurements of laboratory air at split ratios 3, 4, and 5 with sample loop 2 mL**

| | | | Sample loop 2 mL | | |
|---|---|---|---|---|---|
| Gas | Conc. | | Split | | |
| | | | 3 | 4 | 5 |
| | | Repetitions | n=20 | n=20 | n=20 |
| $CH_4$ | 1.895 ppm | Area | 27007 | 24994 | 23355 |
| | | SD | 919 | 1047 | 957 |
| | | CV [%] | 3.40 | 4.19 | 4.10 |
| $CO_2$ | 411 ppm | Area | 8297069 | 6787524 | 5351342 |
| | | SD | 190315 | 89672 | 97529 |
| | | CV [%] | 2.29 | 1.31 | 1.82 |
| $N_2O$ | 339 ppb | Area | 5736 | 4479 | 3359 |
| | | SD | 232 | 139 | 111 |
| | | CV [%] | 4.04 | 3.11 | 3.31 |
| $O_2$ | 20.946% | Area | 4466737 | 3319922 | 2493245 |
| | | SD | 6357 | 4415 | 4366 |
| | | CV [%] | 0.14 | 0.13 | 0.17 |
| $N_2$ | 78.084% | Area | 17054807 | 12671953 | 9517713 |
| | | SD | 23464 | 15061 | 15702 |
| | | CV [%] | 0.14 | 0.12 | 0.16 |

## 3.5 Experimental measurement series comparing the GC results with the reference method (Picarro analyser)

To verify the long-term stability of the measurements and the system performance for real samples, a 55-hour-long measurement series of ambient laboratory air was performed. The subsequent air samples were measured in parallel with the GC setup and with the optical instruments dedicated to analyses of GHG concentrations and isotopic signatures (Picarro G5131-i for isotopic $N_2O$ and Picarro G2201-i for $CO_2$ and $CH_4$) (Picarro, Santa Clara, USA). The reference methods were applied to check whether slight changes in GHG concentrations over the day/night period can be monitored well with this GC system. The reference instruments – isotopic Picarro – showed a quite narrow range of possible concentration measurements (Picarro G5131-i isotopic $N_2O$ up to 2000 ppb, Picarro G2201-i up to 2000 ppm $CO_2$ and up to 12 ppm $CH_4$) but a very high precision for ambient concentrations, without the need for calibration. Therefore they served here as an ideal reference method. During this 55-hour time series, the Picarro measurement was performed every three minutes and the GC measurement every 19 minutes. For GC measurements, split 4 was applied. The concentration trends for $CO_2$ and $CH_4$ were observed to be generally in good agreement and the $N_2O$ concentration was very stable (Fig. 6). Importantly, it was observed for $CO_2$ that after 24 hours the measurements were slightly recalibrated and shifted in relation to the reference method. This indicates the

need for repeated calibration at least every 24 hours, especially for $CO_2$. However, even without recalibration, the maximum difference between the GC measurement and the reference value was below 3 % for both $CO_2$ and $CH_4$. The largest variations in the GC results were observed for $N_2O$, especially when comparing them to the very stable Picarro measurements. This is the most challenging analysis, since $N_2O$ ambient concentrations are lowest and hardest to measure correctly. The maximum difference between the GC measurement and the reference value for $N_2O$ was around 8 % and the standard deviation of GC

measurements was 15 ppb, which represents a 4.4 % error. This is quite high when compared with Picarro statistics where the standard deviation over 55 hours of measurement was 0.24 ppb, which represents less than 0.1 % error. However, for typical $N_2O$ measurements of unknown sample with GC techniques, a 5 % error is a satisfactory result, typically given as an accepted GC measurement error in research studies (Arnold et al., 2001; Harvey et al., 2020). The precision obtained for ambient air measurements is similar for $CO_2$ and $CH_4$ compared with classical FID measurements, with an error of around 2 % (Loftfield

et al., 1997), but is lower when compared with ECD measurements, for which a 1.2 % error has been reported (Loftfield et al., 1997). Usually systems that enable the simultaneous measurement of $CH_4$ and $CO_2$ are very accurate, but the main limitation is the upper detection limit. For example, Wang and Wang (2003) achieved a $CH_4$ precision error in a range from 3.37 % (ambient concentration) to 0.05 % (60 ppm), or for $CO_2$ from 0.66 % (ambient concentration) to 0.04 % (4000 ppm). In the present system, higher concentrations of $CH_4$ can be measured by the TCD simply using a RT-Msieve 5A column. The

chromatographic system can be improved by the addition of a second gas valve aimed at transferring $CO_2$ to the TCD after separation using the Carboxen column. This would allow further improvement of this chromatographic system for the maximum range of measured GHG concentrations.

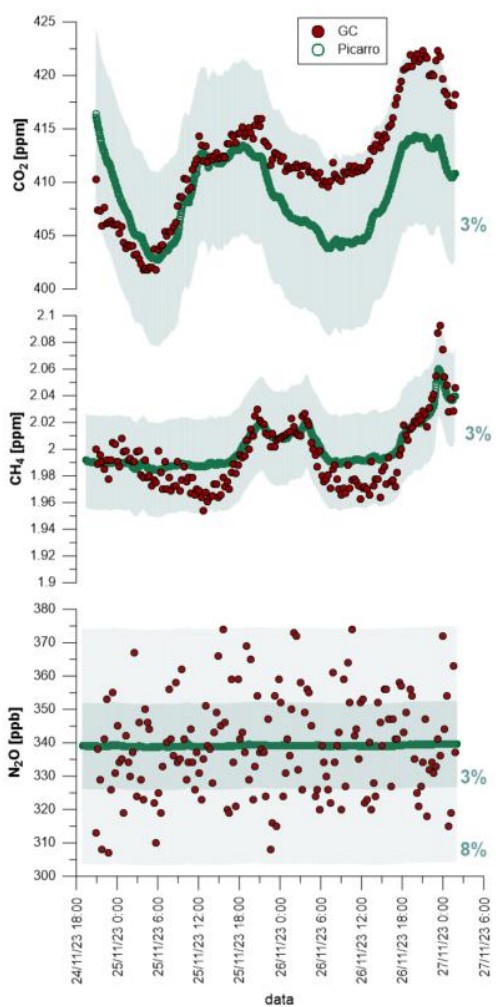

 **Figure 6: Comparison of CH₄, CO₂ and N₂O variations in ambient air measured using a Picarro analyser and the GC system**

## 4. Conclusions

This article outlined a simple method for determination of $CH_4$, $CO_2$ and $N_2O$ concentrations from ambient air. The main advantage is that the use of at least two separate detectors can be avoided, including the ECD that contains radioactive materials,. and FID with methaniser to measure $CO_2$. A single column Carboxen 1010 PLOT can be successfully used for separation of GHG ($CH_4$, $CO_2$, $N_2O$) in time interval enabling measurement of each gas separately without the effect of peak overlapping. In parallel a connected Rt-Msieve 5A column allows for determination of higher $CH_4$ concentrations as well control of the $O_2$ and $N_2$ concentrations in the sample. The main disadvantage of the method is the lack of direct measurement of samples with high $CO_2$ concentrations (above 4000 ppm) in the set-up presented here, which is the upper detection limit for the BID. In summary the detection limits of simplified GC system are:  $CH_4$ 1.8 ppm-100%, $CO_2$ 411-4000 ppm, $N_2O$ 339-

4000 ppm, $O_2$ 0.2-100%, $N_2$ 0.2-100%. Further reconstruction with an additional valve directing the separated $CO_2$ to the TCD would allow additional analyses of higher $CO_2$ concentrations.

Based on performed tests, it is recommended that atmospheric GHGs are analysed using a BID at split ratio 4 or 5 and with a sample loop of 2 mL volume. This would help avoid unnecessary contamination of the Carboxen column with water vapour, therefore splits 1-3 should not be considered for the measurement of environmental gas samples. In this chromatography system, the CV of $N_2O$ measurement at atmospheric level was 11-15 % (sample loop 1 mL), and around 7-9 % (when using the 2 mL sample loop), CV of $CH_4$ measurement at atmospheric level was near 7 % (sample loop 1 mL), and below 3 % (when using the 2 mL sample loop), CV of $CO_2$ measurement at atmospheric level was near 2 % (sample loop 1 mL), and around 1.5 % (when using the 2 mL sample loop). The diurnal variations for $CO_2$ and $CH_4$ can be monitored well with the precision below 3% error, whereas for $N_2O$ measurements 8% error must be taken into account.

The presented results for the measurement precision are satisfactory for the most analytical needs for determining GHG fluxes in field studies or laboratory incubation experiments. However, this GC system is not designed for the most precise analyses at ambient concentration and monitoring daily changes or long-term periods. Its measurement accuracy is not sufficient for such purposes when compared with instruments using optical techniques or automated GC systems dedicated for GHG at ambient air concentration. The greatest advantage of proposed GC system is the ability to measure GHGs at the widest possible concentration, from the near-ambient concentrations to 100%, and in any sample matrix, without the risk of damaging or decalibration of the equipment. It is easy to use and relatively cheap. Therefore, it can be successfully used for analyses of the gas samples with unknown GHG concentrations e.g. from soil chamber measurements, laboratory incubation studies, biogas plants, waste dumps, burning coal heaps, mines, or monitoring of environmental GHGs fluxes.

**Authors contributions:**

MB and PW constructed the analytical set-up, MB and DLS planned the measurement campaign; MB, DLS and PW performed the measurements; MB and DLS analyzed the data; MB, DLS, and PW wrote the manuscript draft; MB, DLS and PW reviewed and edited the manuscript..

**Competing interests:**

The authors declare that they have no conflict of interest.

**Funding:**

This study was financially supported by the "Polish Returns" programme of the Polish National Agency of Academic Exchange and the grant Opus No. 2021/41/B/ST10/01045 funded of the National Science Centre Poland.

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
