# Peer review of "Simultaneous measurement of greenhouse gases (CH4, CO2 and $N_2O$ ) using a simplified gas chromatography system"

_EGUsphere, 2024_

## Referee Comment (RC1)

[referee-annotated manuscript omitted]

---

## Author Comment (AC1)

**Response to Reviewer #1**

Thank you very much for your positive evaluation on our manuscript and the critical comments which helped us to prepare the improved version of our work.

Please find below our responses and clarifications (black font) and the proposed changes that will be made in the manuscript (blue font).

Page 3, line 89

*What SRI stand for? please write first*

This is the name of the producer, we will clarify this:

AS-210 Greenhouse Gas Autosampler (SRI Instruments Europe GmbH, Bad Honnef, Germany)

Page 3, line 90

*Molecular sieve*

Yes, the own name of the column was used. This sentence will be corrected with precise information :

The GC separation columns used in this study were performed with a porous layer open tubular column (Carboxen 1010 PLOT) and a molecular sieve column (RT-MSieve 5A), which assured the full separation of the analysed gases.

Page 3, line 95

*The schematic configuration of GC system for simultaneously measurement of CO2, CH4, and N2O should be provided to get better understanding of the this measurement*

Thank you for this suggestion, this will be definitely helpful for the readers. The following scheme will be added to Section 2.1, as Figure 1

[Figure]

Page 3, line 115

*The statement of high sensitivity of BID and the usage of discharge flow, is there any references? (please cited in this statement)*

This is information from Shimadzu Instruction Manual. The respective citation will be added: Barrier Discharge Ionization Detector for GC-2010 Plus BID-2010 Plus Instruction Manual. Shimadzu Corporation 2013

Page 4,line 99

*Liquid N2 (LN2)*

It will be corrected.

Page 4, line 107 and 108

*Column has length 30 m, 0.32 mm inner diameter (i.d). Please provide also the information of thickness of stationery phase?*

This missing information will be added. Sentence will be corrected as below:

Additionally, using a t-joint, the injection sample was then divided between two porous layer open tubular capillary columns filled molecular sieve 5A (RT-MSieve 5A 30 m x 0.32 mm x 50 µm, Restek, USA, #18284) and fused silica (Carboxen 1010 PLOT 30 m x 0.53 mm x 0.15 µm, Supelco, USA, #24246).

Page 4, line 113

*Porous-layer open-tubular?*

Yes, phrase will be corrected to "the porous layer open tubular (PLOT) capillary column"

Page 4, line 124

*Please express with the consistence significant figures of the mL value (two digit or one digit behind coma)*

The values will be corrected to one digit behind coma.

Page 4, line 128

*the standard gas mixtures used in this study, is the certified standard gas mixtures or in-house standard gas mixtures developed by your institute?*

We used in-house standard of compressed air from Wrocław (Poland), which contained natural moisture (vapour) in the 10 liters gas cylinder. The second standard was ordered in Messer Polska Sp. z o.o. The special gas mixture is the commercially product prepared in Switzerland according to the norm ISO6141:2015. This standard was prepared in pure $N_2$, without moisture, in volume of 8 liters. Based on information from Messer each of the standard contain filter F10, which protects outer valve to possible vapour or solid particles.

This information will be added to standards description.

Page 4, line 129:

Concentrations are the same as showed in the second columns in the Tables 1 and 2. The link to the values in respective tables will be added (Exact values in Table 1 and 2).

| Gas | Concentration [% or ppm] |
|-----|--------------------------|
| $CH_4$ | 1.895 ppm |
| $CO_2$ | 411 ppm |
| $N_2O$ | 0.339 ppm |
| $O_2$ | 20.946 % |
| $N_2$ | 78.084 % |

Page 5, line 149

*The peak label (name) for identification of gas component cannot be seen clearly (too small) both in the picture and inset of picture*

The previous version contained the picture obtained using GC Software Lab Solutions dedicated for GC made by Shimadzu. In the response for Reviewer's comments the data of chromatograms were exported from Lab Solutions as ASCII file and therefore converted to prepare new plots using Grapher Golden Software. Here we present the new Figure 2A and 2B (zoomed Figure 2A) which will be added to the revised article.

Figure 2A – presented below

[Figure]

Figure 2B – zoomed figure 2A

[Figure]

Page 6, line 163

*The peak label (name) for identification of gas component cannot be seen clearly (too small) both in the picture and inset of picture*

New Figure 3 will be corrected as below:

[Figure]

Page 6, line 164

*Is the compressed air standard mixtures contained the moisture (dry or wet compressed air standard mixtures).*

*How the compressed air standard mixtures prepared should be explained in Materials and method section?*

The compressed air contained natural moisture (water vapour). We analysed standard with natural moisture, because every sample collected form the natural conditions (field, soil, studies) has natural moisture (vapour). Our method is dedicated for such type of samples. This will be clarified in compressed air description.

Page 6, line 178

*In this section, can $O_2$ and $N_2$ be separated well by the GC system and detected by TCD? because in section 3.1, the $O_2$ and $N_2$ peaks overlapped when analyzed with GC BID as seen in the chromatogram in Figure 2.*

*Are the conditions and setup of the GC system in sections 3.1 and 3.2 different?*

Yes, in all configurations the system allow for separation of $O_2$ and $N_2$ by molecular sieve 5 A connected to TCD. The chromatogram with separation using this column will be added to manuscript as below:

[Figure]

The sections 3.1 contains chromatogram for tested configuration with using of the column SH-Q-BOND. The GC setup for preliminary tests of the SH-Q-BOND column connected to BID detector was different than GC setup for using Carboxen 1010 PLOT. Configuration of system with Carboxen 1010 PLOT was in details showed in the section 2.1, and was used to obtain results described in section 3.1, 3.2, 3.3, 3.4 and 3.5. We didn't describe the details of configuration with SH-Q-BOND column in the submission, because we didn't present any results regarding the accuracy of measurements using SH-Q-BOND column. On Figure 1 we showed example of the chromatogram (Figure 1) from SH-Q-BOND column and briefly described the analytical problems related to the separation of $CO_2$ and $N_2O$ and the lack of visibility of $CH_4$ in atmospheric concentration in the text. Conditions and setup of testing with SH-Q-BOND were exactly the same (injector, TCD, BID configuration, presence of MSieve 5A, the splitting), the only difference was oven temperature programme which was as below:

| Rate | Temperature | Hold Time |
|---|---|---|
| - | 35.0 | 3.50 |
| 10.00 | 130.0 | 0.50 |

We will add basic information of the setup with SH-Q-BOND as well specific conditions used during testing as an attachment Supplementary Materials.

*The separation chromatogram for analysis in the GC system in section 3.2 can be displayed*

Yes, $N_2$ and $O_2$ in our GC system can be separated with application of RT-MSieve 5A column with detection using TCD. The range of detection is from 0.2 to 100% (for $N_2$ and $O_2$). This is very important especially for analyses of microbial samples (cultivation experiments) to control the aerobic or anaerobic conditions in the headspace gas. Moreover, in our system we have potential to separate argon from ambient air using cryogenic option (decrease of the column oven temperature to e.g. 0°C by liquid nitrogen ($LN_2$)), because RT-MSieve 5A column can work in temperature range of -100 to 300°C. However, if it is necessary to perform an analysis of $O_2$ concentration at lower concentration (below 0.2%) and detect with BID, Carboxen 1010 PLOT column should be unmounted from the GC system – because its minimal temperature of working is 25°C. It is also possible to separate $N_2$ from $O_2$ using SH-Q-Bond column which can work at minimal temperature of -60°C.

Page 7, line 194

*For the better understanding, the data can be added with split ratio of 1,2 at 2 mL sample loops*

We decided not to do it, because the amount of water vapour which can be transferred through the column after injection of 2 ml of ambient air sample would be destructive for the column's filling phase (will shorten its proper functioning). Material used for Carboxen columns is known as very sensitive for water vapour – in the past its filling was used as water adsorber – please see:

Fastyn P., et al. 2003: Adsorption of water vapour from humid air in carbon molecular sieves: Carbosieve S-III and Carboxens 569, 1000 and 1001 - Analyst (RSC Publishing)

Page 7, line 189

*The detection limit of simultaneous analysis $N_2O$, $CH_4$, $CO_2$ using GC BID can be provided further to get the information of the characteristics of this GC system*

In case of $CH_4$ and $N_2O$ we are working at very low levels for BID, almost at the limit of detectability. For $CH_4$ it is 1.8 ppm and for $N_2O$ 339 ppb. The precision error of $N_2O$ at such low concentration is 5%. We are aware that this precision is not really satisfactory for measuring slight variations of ambient levels, however can be very well applied to determine environmental fluxes. We are still working on improvements, and we will test few more modifications, e.g. installing additional valve to allow for more gas measurements combinations and temperature controlling of sample loop and injection valves to better control the water vapour level. With this manuscript we intended to publish the first idea of the system with Carboxen column and BID detector already applicable for many environmental studies. Hopefully in near future we can report better precision.

Page 7, line 194

*Why the special gas mixture used in this analysis do not contain the same matrices as compressed air or ambient air (it means the $CH_4$, $CO_2$, and $N_2O$ is in air matrices)*

This reason is the process of preparing the calibration mixture by the manufacturer. To obtain the appropriate accuracy, the manufacturer uses pure gas that doesn't contain the analyte, which needs to be added and diluted for final concentration.

Our aim was to use the GC system also for samples from laboratory incubation studies with partially anoxic conditions. For such experiments it is important to determine the low oxygen levels, therefore we needed a gas standards with no and low $O_2$ concentration. To reduce the amount of necessary standards tanks we simultaneously varied $O_2/N_2$ levels and levels of GHGs possibly widest range of concentrations of all gases with lowest amount of tanks.

Page 10, line 238

*How about the comparison of precision from this GC system and CRDS Picarro in $CO_2$ and $CH_4$ measurement (standard deviation, %relative standard deviation, error)?*

For these comparison measurements we could only determine % error, because we measured atmospheric ambient air and it shows natural diurnal variations, we compared the results point by point and not the means and standard deviations. This could be only done this way because CRDS Picarro is dedicated mostly to ambient air samples, and it is not possible to insert special gas samples.

Page 10, line 248

*Regarding the upper detection limit, How much the upper detection limit of this simultaneous measurement of $CH_4$, $CO_2$, $N_2O$ by the GC system in this study?*

We can state at this moment that concentrations which can be measured using our GC system are:

$CH_4$ 1.8 ppm to 4000 ppm (using BID), and 0.2% to 100% (using TCD)

$CO_2$ 411 ppm to 4000 ppm (0.4%) – according to Shimadzu specifications of BID detector

$N_2O$ 0.330 ppm (339 ppb) to 4000 ppm – according to Shimadzu specifications of BID detector

Additional information:

$O_2$ – 0.2 to 100% (using TCD)

$N_2$ – 0.2 to 100% (using TCD)

---

## Author Comment (AC2)

**Response to Reviewer #2**

*General comments*

*This paper reports a method for analyzing three greenhouse gases simultaneously using a relatively simple GC system. The strength of this paper is that the developed method enables us to separate and quantify CH4, CO2, and N2O in air samples on a single column with a single detector. This technique would reduce sample size, time, and resources for the gas analyses. A shortcoming of this paper is that the precision or repeatability of the method is insufficient for atmospheric monitoring at background concentration level. In this regard, I think the title is misleading and should be revised to mean that the method is most suitable for source gases such as soil emission. Another concern is that the results of experiments for optimizing the GC setting are mainly discussed in the context of CV without further consideration of sensitivity of the detector. I believe combination of split ratio and sample size affect the amount/concentration of the target species delivered to the detector. For example, split ratio of 1 with 1-mL sample loop should give peak area that is equal to the area obtained at split ratio of 3 with 2-mL loop. In Table 1, I see results of CO2 and N2 are consistent with this idea, but it is not the case for other gases.*

*In summary, I recommend the publication of this paper after the authors address the issues above and specific points below.*

Thank you very much for your positive evaluation on our manuscript and the critical comments which helped us to prepare the improved version of our work. Yes, definitely we proposed inappropriate title for this manuscript, this will be changed to: ″**Simultaneous measurement of greenhouse gases ($CH_4$, $CO_2$ and $N_2O$) using a gas chromatography system".** Our initial idea was to test this system for ambient air measurements and although so far the precision is not sufficient for precisely measure the small atmospheric variations, it can be very well applied for measurements of atmospheric fluxes. We missed to review our title before the final submission.

The answer for questions regarding the sensitivity of the BID detector is based on the analysing S/N (signal/noise) ratio for the selected peak. Generally, S/N above 3 allows for identification of the peak, whereas S/N above 10 allows for quantitative determination of concentration. For 339 ppb $N_2O$ analysis at split ratio 5 we achieved the ratio S/N usually between 12-15. The S/N ratio for 50 ppm $N_2O$ standard was usually above 250. The method how we calculated SD and CV [%] is described in the detailed answers to Reviewer's comments. We presented the equations and method of obtaining the CV [%] for this data. The very high peak area of $O_2$ (Table 1, sample loop 2 ml, split 1) was incorrect (mistake during preparation of the final table) and the corrected value 4513907 will be inserted in the table, which is similar to value 4317080 obtained with split 1 and sample loop 1.

Generally, transferring of the sample from the columns should be very fast to avoid the flattening of the peaks. The sample is transported from the sample loop to the column at total flow speed. Total flow is mainly dependent on the split value: total flow = column flow + split ratio x column flow + purge flow (3ml/min). Therefore, the larger the split value, the faster the sample reaches the column. The narrower band in which the sample hits the column makes the peaks narrower and higher than with lower split values, even though less sample hits the column. However, this is one of the rules that must be checked experimentally each time to find appropriate values for the experiment being conducted.

In this method we also consider the amount of water vapour, which is transferred to the column. It is important to find the ideal compromise between the appropriate amount of gas supplied to

the column and obtaining a strong signal, especially for the lowest concentrations using BID. Therefore, we assume that the peak area are not consistent with idea of the Reviewer mainly due to the difference in the speed of transferring through the columns. The additional calculation is showed below to compare all uncertainties. The difference between obtained peak areas is showed in the table as column C, and its absolute value in column D. Then in column E we showed that the value A (difference of the peak area expressed in %) is much higher for smaller peaks ($CH_4$ and $N_2O$, 23.30 and 28.66%, respectively). For larger peaks of $CO_2$, $O_2$ and $N_2$ the difference E is equalled around 5%. Therefore, it is very important to monitor parameter of S/N ratio when analysing low concentration samples.

| Gas | A
Peak area at split 1 sample loop 1mL | B
Peak area at split 3 sample loop 2mL | C
C=A-B | D
Absolute value of C | E [%]
E=(A*100)/D |
|---|---|---|---|---|---|
| $CH_4$ | 33294 | 25535 | 7759 | 7759 | 23.30 |
| $CO_2$ | 10900323 | 10274295 | 626028 | 626028 | 5.74 |
| $N_2O$ | 7801 | 5565 | 2236 | 2236 | 28.66 |
| $O_2$ | 4317080 | 4513507 | -196427 | 196427 | 4.55 |
| $N_2$ | 16522678 | 17338000 | -815322 | 815322 | 4.93 |

Please find below our responses to the specific points and clarifications (black font) and the proposed changes that will be made in the manuscript (blue font).

*Specific comments*

*L40–41. I think "the most modern techniques and devices" should be adopted only if they provide results with precision and accuracy that are sufficient for the purpose.*

We agree with this comment and we are aware it is not suitable for measuring minimal changes in greenhouse gas concentrations in ambient air over time (and such measurements are required for monitoring climate change). The advantage of our method is that it determines measurements in a wide range of concentrations (up to 4000 ppm, according to specifications of BID). Therefore, we will change the title, which in the original version is misleading and confusing for the readers. In the corrected version of the article, we will not specify limitations of the method only for ambient gases. Our aim is to show novel method of separation $CH_4$, $CO_2$ and $N_2O$ with single detector and one column.

*L46. I think a chapter from an e-book is referred here. Correct the citation information in the References section.*

You have right. This is a chapter from the e-book. Citation will be corrected:

Zaman, M., Kleineidam, K., Bakken, L., Berendt, J., Bracken, C., Butterbach-Bahl, K., Cai, Z., Chang, S.X., Clough, T., Dawar, K., Ding, W.X., Dörsch, P., dos Reis Martins, M., Eckhardt, C., Fiedler, S., Frosch, T., Goopy, J., Görres, C.-M., Gupta, A., Henjes, S., Hofmann, M.E.G., Horn, M.A., Jahangir, M.M.R., Jansen-Willems, A., Lenhart, K., Heng, L., Lewicka-Szczebak, D., Lucic, G.,

Merbold, L., Mohn, J., Molstad, L., Moser, G., Murphy, P., Sanz-Cobena, A., Šimek, M., Urquiaga, S., Well, R., Wrage-Mönnig, N., Zaman, S., Zhang, J., Müller. C., 2021. Methodology for Measuring Greenhouse Gas Emissions from Agricultural Soils using Non-Isotope techniques. 11-209. In: Zaman M, Kleineidam K, Bakken L, Berendt J, Bracken C, Butterbach-Bahl K, Cai Z, Chang SX, Clough T, Dawar K, Ding WX, Dörsch P, dos Reis Martins M, Eckhardt C, Fiedler S, Frosch T, Goopy J, Görres C-M, Gupta A, Henjes S, Hofmann MEG, Horn MA, Jahangir MMR, Jansen-Willems A, Lenhart K, Heng L, Lewicka-Szczebak D, Lucic G, Merbold L, Mohn J, Molstad L, Moser G, Murphy P, Sanz-Cobena A, Šimek M, Urquiaga S, Well R, Wrage-Mönnig N, Zaman S, Zhang J, Müller C (2021) Measuring Emission of Agricultural Greenhouse Gases and Developing Mitigation Options Using Nuclear and Related Techniques Springer ISBN 978-3-030-55395-1, https://doi.org/10.1007/978-3-030- 55396-8.

*L53–55. I think mass spectrometer is one of the detectors used in gas chromatography. I cannot understand why the mass spectrometry is specifically noted.*

You have right, we will delete the part of the sentence "gas chromatography and/or gas chromatography coupled with mass spectrometry (GC-MS)". Corrected version will be as below:

Thus the most reliable methods for checking samples in a very wide range of concentrations are chromatographic methods (Ekeberg et al., 2004).

*L56–63. This paragraph is difficult to read because there is a lot of duplication. It seems that the authors try to list several types of "systems", but they just mention detectors and GC columns.*

You have right. We will correct the paragraph to avoid duplication. The new paragraph will be:

Gas chromatography with automated sampling injections is very common and user friendly. The most common GHG measurement systems have been developed with: a thermal conductivity detector (TCD) (measurement of $CH_4$ and $CO_2$), flame ionisation detector - FID (measurement of $CH_4$ and $CO_2$ using a methaniser), electron capture detector - ECD (for $N_2O$ measurement) (Hedley et al., 2006; Loftfield et al., 1997; ; Wang and Wang, 2003).

*L64. What does "natural wear" mean? Because the half-life of the radiation source Ni-63 is 100 years, I wonder other factors are meant.*

The ECD detector cell is a consumable part. The cells should be replaced every 2-5 years. Moreover, the ECD is very sensitive to oxygen, which oxidizes the nickel foil. Using poor quality nitrogen, e.g. 5.0, which contains trace amounts of oxygen, is enough to shorten the cell's life. Before the ECD, oxygen traps are used that must be replaced regularly. In addition, the ECD gets dirty with the stationary phase from the column. In general, the thicker the film inside the chromatography column, the faster the cell wears out. The ECD also gets dirty simply with the measured analytes. The ECD cell can no longer be disassembled and cleaned. Theoretically, it can be sent to service for cleaning, but the price is so prohibitive that it is not worth doing (the price of cleaning ~ the price of buying a new cell). The ECD can only be annealed. On the other hand, the BID is a maintenance-free detector, by definition it does not get dirty due to the dielectric barrier.

*L78. It is not clear what "and/or" means. In a certain case, both the two-column system and the single- column system are required?*

You have right, this I misleading phrase. We will delete "and/or" and leave only word "or".

*L81–83. This sentence is difficult to understand. For example, what is compared using "as well as"?*

We will correct this sentence (confusing part of the sentence will be deleted). The corrected version of the sentence is:

This set-up using single column and single BID detector is commonly used for determination of $CH_4$ and $CO_2$ at very low atmospheric concentrations (Gruca-Rokosz et al., 2020).

*L95. I recommend to add a schematic figure showing the GC system.*

Thank you, this is a very good idea. We will add such scheme of the system to the revised manuscript.

[Figure]

*L101–102. "warmed" at what temperature?*

It was 110°C. This information will be added to the text and is showed on scheme of the GC system.

*L102–103. This means gas sample with high moisture (e.g., soil gas) might cause problems. Is the system designed for already dried samples or does it withstand moist samples?*

We tested Messer standard which doesn't contain water vapour as well environmental samples with natural, high moisture (collected from the soil after rainfall etc.). Generally for samples with significantly higher concentrations of $CH_4$, $CO_2$ and $N_2O$ presence of moisture does not significantly affect the measurement. However, for samples with ambient concentrations of $N_2O$ and with high moisture the baseline (and noise) should be checked to be sure that the signal of $N_2O$ is enough to measure the concentrations. Shimadzu recommends that S/N (signal/noise) ratio should be above 10 to be sure that determination is correct. For 339 ppb $N_2O$ analysis at split ratio 5 we achieved the ratio S/N usually between 12-15. The S/N ratio for 50 ppm $N_2O$ standard was usually above 250.

The analysis of gas samples with natural moisture will lead to faster decay of the column filling (film) and accumulation of the solid parts in the particle trap, as a result, to a rise of the baseline, which has the greatest impact on the determination of samples with extremely low concentrations. For prevention we are actually testing different moisture traps (connected to ta

capillary between GC and autosampler). This is to minimize the presence of water vapour inside the system (such as the sample loop).

*L107–110. I guess the length and inner diameter of the columns are described here, but the splitting ratio of 1:5 cannot be achieved with the dimensions shown here. Add other parameters such as thickness of the inner coating of the columns. Also, combination of column and detector should be clearly described or shown using a figure. Is the TCD connected to the molecular sieve column?*

We will modify the description in the text as well showed basic information on GC scheme. The corrected text will be as below:

Additionally, using a t-joint, the injection sample was then divided between two porous layer open tubular capillary columns filled molecular sieve 5A (RT-MSieve 5A 30 m x 0.32 mm x 50 µm, Restek, USA, #18284) and fused silica (Carboxen 1010 PLOT 30 m x 0.53 mm x 0.15 µm, Supleco, USA, #24246). The dimensions of the columns were selected to achieve a splitting ratio of 1:5, directing most of the sample to the Carboxen 1010 PLOT and BID. Corresponding calculations were performed in Shimadzu AFT (Advanced Flow Technology) software.

*L111. Quantitative information should be given instead of "extremely low baseline noise".*

For example the baseline noise for measurement of 339 ppb $N_2O$ with split 5 is 27.53 and S/N is equalled 11.65.

*L115–117. It is not clear at which position the discharge gas is added to the flow system.*

Discharge gas is connected and passed from the top, it is used to create plasma, below is a technical scheme from Shimadzu manual instruction:

[Figure]

Please see also link to the Schimadzu website BID | Research & Development | SHIMADZU CORPORATION with details regarding BID.

*L117. I cannot understand what this sentence means.*

On the TCD channel satisfactory sensitivity was achieved with standard settings - current 80 mA value and make-up gas 8 mL/min. Generally, in TCD, sensitivity can be adjusted by changing the type of carrier gas, make-up and current.

This will be clarified in the manuscript.

*L121–122. Do the authors mean the final temperature of 200C is kept for 1 min? Revise the sentence.*

Yes. The final temperature is called "hold time". The temperature program is also now showed on the figure with GC Scheme.

*L123–124. As described in the previous section, the flow after sample injection was divided into two columns. Are these flow parameters common to them?*

It is necessary to add 2.5m particle traps with a diameter of 0.32mm to the columns. For a column with a diameter of 0.32mm, we added 2.5m which gives 32.5m. The column of 0.53mm should be theoretically extended by 19m, therefore 2.5m x 0.32mm gives the same resistance as 19m x 0.53mm, therefore the second column has a entered length of 49m. The flow parameters are very close to those showed on picture:

This will be added in the manuscript:

[Figure]

*L140. Decrease from what temperature?*

What we mean here is a quick stabilization of the oven temperature. Temperature stabilization. 35° is the lowest temperature that can be practically achieved in our laboratory without using liquid nitrogen for cooling. The temperature reduction that we mention here concerned the standard temperature of 40°C (at which we most often tested the SH-Q-Bond column).

This will be clarified in the manuscript.

*Figures 1 and 2. Labels on the x and y axis are difficult to read.*

The Figures are corrected with larger size fonts.

*L158. Does "vapour" mean water vapor? Please specify. Also, do the authors mean that the retention time of H2O peak shown in Figure 2 changes depending on the amount of water in the sample?*

Yes, vapour means water vapour (natural moisture). The $H_2O$ peak appeared usually when the oven temperature reached 115°C – it is not perfectly visible because the oven temperature was rising (and thus the baseline level too).

*L165. It is not clear to me for what purpose the authors made experiments with different combinations of split ratio and sample size. I think the amount of sample (and water vapor) injected to the column is determined by the two parameters. For example, if 1 mL sample is processed with split ratio of 1, the amount of sample injected to the column is 1×1/(1+1) = 0.5 mL. If 2 mL sample is processed with split ratio of 3, the amount would be 2×1/(1+3) = 0.5 mL, which is the same as the first case.*

We tested different split settings and loop volumes to find a compromise between the amount of sample analyzed and the speed of sample transport to the chromatography column. If we use a 2 mL loop, we can dose a larger amount of sample onto the column, which we want to transport to the column as quickly as possible. If we do it slowly, then the peaks will be broad and low, we will lose sensitivity.

*L177 and elsewhere. Since TCD, FID, and ECD are acronyms of "xxx detector", notation like "TCD detector" is awkward.*

Thank you for this comment. We will correct these mistakes.

*Table 1. It seems "SD" does not show the standard deviation of peak area, because dividing this value with "area" gives much smaller CV value. This is also the case for Tables 2 and 3. Please correct.*

You have right. We suppose that these ambiguities resulted from the different way for presentation of the data (SD and CV) and lack of a clear presentation of the used calculation method in the table's legends. In the original manuscript SD showed the standard deviation of the calculated concentration. First, we measured the same standard 3, 4, 5, 10 or 20 times. Then, this raw data of measured peak area was used for calculation of mean peak area typical for concentration at different and known levels (calibration). Concentrations were calculated according to equation:

Concentration [%] = (measured peak area * known concentration of the standard) / (calculated mean peak area from 3, 4, 5, 10 or 20 analyses)

SD = standard deviation of concentrations calculated for 3, 4, 5, 10 or 20 analyses

CV = (SD * 100) / (mean peak area from 3, 4, 5, 10 or 20 analyses)

However, we corrected tables according to the idea proposed by the Reviewer. Tables 1, 2 and 3 after correction and re-calculation to present SD of the measured area are at the end of the response to Reviewer 2. We will add clarification regarding calculations (SD, CV) used for this study at the bottom of each table. We will also modify sentences, where the range of CV needs to be revised (line 169- phrase "and ranged from 0.57 to 1.87 % for the 2 mL sample loop" and line 170 – phrase "and ranged from 7.09 to 12.67 %, compared with the 1 mL sample loop"). These changes do not affect the conclusions from the obtained results.

*L185. Specify "the main atmospheric gases".*

We mean $O_2$ and $N_2$. We will correct the paragraph because the measurement of these two compounds was performed with TCD – in our system is dedicated to a gases of higher concentrations than 0.2% (2000 ppm). Therefore the comparison of the SD obtained from BID which is much more sensitive in our opinion it is not recommended. New paragraph will be as below:

The gases analysed using the TCD detector, $O_2$ and $N_2$, were characterised by a narrow CV ranging from 0.10 to 0.39 %. The highest CV (0.39 %) was observed for the $N_2$ measurement with the sample loop 2 mL, where the peak area was very large. The results of the measurement (peak area, SD, CV) are presented in Table 1.

*L202. What does "reportable results" mean?*

This was a mistake, most probably made by autocorrection. We should use the phrase "repeatable results" – the achieved precision of $N_2O$ analysis is considered as sufficient for determining environmental gas samples, e.g. from soil experiments and measurements of gas emissions from natural sources such as peat bogs.

*L212–214. If the authors use the system for soil gases, should the sample be dried before analysis?*

No, we recommend to use original sample and eventually trap the water moisture (using sorbents) – but these sorbents needs to be tested in order to check potential influence on the concentrations of other gases (e.g. sorption of $CO_2$ or $N_2O$). The best way is to analyse pure samples, and find the conditions when water moisture does not affect the precision of GC measurements.

*Table 3. Is the area for N2O at split ratio of 3 correct? It is extremely higher than those obtained at split ratio of 4 and 5.*

Thank you for this comment. This is mistake. We will correct the table. The correct value of area= 5552, SD=30.6, CV [%]=9.02.

*L234 split ratio of 4*

It will be corrected.

*L235–237. I cannot understand these sentences. Do the authors mean the "GC" results of CO2 obtained after 25/11/23 18:00 (Figure 3) have a systematic error due to a shift in sensitivity of the detector?*

Yes, some shift occurred, eg. due to change in air moisture, this was Friday evening after all people left laboratory. But importantly, with new calibration the both measurements would stay in very good agreement.

One of the reason why Picarro is much more precise could be amount of sample analysed. Picarro analyses at least 20mLof sample for constant measurement, which make a single measurement every few seconds. Our GC system is sampling air in the volume of 1 or 2 mL. By design, such a measurement will not be as accurate in the long-term run as a device specifically dedicated to analyzing $CO_2$ or $N_2O$ fluctuations in the atmosphere. Our system is not limited to low concentrations of GHG's, which is why it stands out from other special devices.

*L243–245. Although it is not clear what "typical N2O measurements" means, 5% error is NOT satisfactory if one would like to know diurnal/seasonal cycle or secular trend of atmospheric N2O at background level.*

Yes, we agree with this comment. However, in soil sciences such error is accepted by scientific community.

*L250–251. Since no TCD chromatogram is shown, this statement cannot be verified.*

We will add the chromatogram with TCD to the manuscript as below:

[Figure]

*L261. Consider other quantitative or scientific expression for "very good".*

Instead of the expression "very good" we will use: "time interval enabling measurement of each gas separately without the effect of peak overlapping"

*L268. water vapour?*

Yes, we will correct the sentence.

**Table 1: Peak area, SD (standard deviation) and CV (coefficient of variation) of standard atmospheric gas measurements at split ratios 1, 2, 3, 4, and 5 with 1 mL and 2 mL sample loops**

| Gas | Conc. | | Sample loop 1 mL | | | | | Sample loop 2 mL | | |
|---|---|---|---|---|---|---|---|---|---|---|
| | | | Split | | | | | Split | | |
| | | | 1 | 2 | 3 | 4 | 5 | 3 | 4 | 5 |
| | | Repetitions | n=3 | n=10 | n=20 | n=20 | n=20 | n=10 | n=10 | n=10 |
| CH$_4$ | 1.895 ppm | Area | 33294 | 24249 | 25271 | 20085 | 16269 | 25535 | 23901 | 23195 |
| | | SD | 2580 | 802 | 1859 | 1365 | 1394 | 386 | 381 | 640 |
| | | CV [%] | 7.75 | 3.31 | 7.36 | 6.79 | 8.57 | 1.51 | 1.59 | 2.76 |
| CO$_2$ | 411 ppm | Area | 10900323 | 8015901 | 5742835 | 4525868 | 3658827 | 10274295 | 8037126 | 6295494 |
| | | SD | 99092 | 84289 | 162281 | 58578 | 79062 | 155997 | 45587 | 117747 |
| | | CV [%] | 0.91 | 1.05 | 2.83 | 1.29 | 2.16 | 1.52 | 0.57 | 1.87 |
| N$_2$O | 0.339 ppm | Area | 7801 | 4732 | 3572 | 2565 | 2080 | 5565 | 4470 | 3306 |
| | | SD | 714 | 955 | 554 | 289 | 323 | 705 | 317 | 329 |
| | | CV [%] | 9.15 | 20.19 | 15.51 | 11.26 | 15.55 | 12.67 | 7.09 | 9.95 |
| O$_2$ | 20.946% | Area | 4317080 | 3030077 | 2147582 | 1660466 | 1340040 | 4513907 | 3320157 | 2523219 |
| | | SD | 9241 | 5136 | 7177 | 4196 | 2088 | 10598 | 3298 | 7454 |
| | | CV [%] | 0.21 | 0.17 | 0.33 | 0.25 | 0.16 | 0.23 | 0.10 | 0.30 |
| N$_2$ | 78.084% | Area | 16522678 | 11587989 | 8205177 | 6343410 | 5119796 | 17338000 | 12767184 | 9709091 |
| | | SD | 42456 | 21960 | 27025 | 16376 | 8251 | 31998 | 12603 | 38248 |
| | | CV [%] | 0.26 | 0.19 | 0.33 | 0.26 | 0.16 | 0.18 | 0.10 | 0.39 |

**Table 2: Peak area, SD (standard deviation) and CV (coefficient of variation) of special gas mixture measurements at split ratios 3, 4, and 5 with sample loops 1 mL and 2 mL**

| Gas | Conc. | | Sample loop 1 mL | | | Sample loop 2 mL | | |
|---|---|---|---|---|---|---|---|---|
| | | | Split | | | Split | | |
| | | | 3 | 4 | 5 | 3 | 4 | 5 |
| | | Repetitions | n=5 | n=4 | n=3 | n=10 | n=10 | n=10 |
| $CH_4$ | 10 ppm | Area | 149922 | 118357 | 96007 | 278296 | 207227 | 161265 |
| | | SD | 831 | 133 | 318 | 1067 | 707 | 2895 |
| | | CV [%] | 0.55 | 0.11 | 0.33 | 0.38 | 0.34 | 1.79 |
| $CO_2$ | 1000 ppm | Area | 13522596 | 11079679 | 8993195 | 21485425 | 17323467 | 13525917 |
| | | SD | 331786 | 341314 | 184053 | 337763 | 331554 | 435439 |
| | | CV [%] | 2.45 | 3.08 | 2.05 | 1.57 | 1.91 | 3.22 |
| $N_2O$ | 50 ppm | Area | 629251 | 497331 | 402025 | 1171624 | 867747 | 673486 |
| | | SD | 2801 | 1782 | 729 | 9975 | 11510 | 14360 |
| | | CV [%] | 0.45 | 0.36 | 0.18 | 0.85 | 1.33 | 2.14 |

**Table 3: Peak area, SD (standard deviation) and CV (coefficient of variation) of direct measurements of laboratory air at split ratios 3, 4, and 5 with sample loop 2 mL**

| Gas | Conc. | | Sample loop 2 mL | | |
|---|---|---|---|---|---|
| | | | Split | | |
| | | | 3 | 4 | 5 |
| | | Repetitions | n=20 | n=20 | n=20 |
| $CH_4$ | 1.895 ppm | Area | 27007 | 24995 | 23355 |
| | | SD | 919 | 1047 | 957 |
| | | CV [%] | 3.40 | 4.19 | 4.10 |
| $CO_2$ | 411 ppm | Area | 8297069 | 6787524 | 5351342 |
| | | SD | 190315 | 89672 | 97529 |
| | | CV [%] | 2.29 | 1.32 | 1.82 |
| $N_2O$ | 339 ppb | Area | 5736 | 4479 | 3359 |
| | | SD | 232 | 139 | 111 |
| | | CV [%] | 4.04 | 3.11 | 3.31 |
| $O_2$ | 20.95% | Area | 4466737 | 3319922 | 2493245 |
| | | SD | 6357 | 4415 | 4366 |
| | | CV [%] | 0.14 | 0.13 | 0.18 |
| $N_2$ | 78.08% | Area | 17054807 | 12671953 | 9517713 |
| | | SD | 23464 | 15061 | 15702 |
| | | CV [%] | 0.14 | 0.12 | 0.16 |

---

## Author Comment (AC3)

**Response to Reviewer #3**

*I am afraid I have to reject this paper in its present form, simply because to my opinion the instrument in its present state lacks being fit-for-purpose in any application.*

*While the simultaneous detection of the three anthropogenic greenhouse gases using just one detector and only He is an improvement compared to previous GC systems, the precision and accuracy is much too poor for any application, most certainly for atmospheric measurements. Compare for instance to van der Laan et al (2009): A single gas chromatograph for accurate atmospheric mixing ratio measurements of CO2, CH4, N2O, SF6 and CO. Atmospheric Measurement Techniques 2, 549–559 (2009). (not cited in the paper). They use two detectors, FID and EC, but they reach precisions : ±0.04 ppm for CO2, ±0.8 ppb for CH4, ±0.8 ppb for CO, ±0.3 ppb for N2O, and ±0.1 ppt for SF6, which are within the reach of the WMO recommendations for atmospheric monitoring. The present instrument gets uncertainties more than two orders of magnitude worse!*

*If the authors have other applications in mind, soil air or agricultural greenhouse air for example (as several of the papers they cite, and which is hinted at by the values for their reference cylinder with strongly enhanced concentrations compared to ambient), then they should make that very clear in the paper. Furthermore, I think that even for these applications the present quality is not good enough.*

*The authors should compare their instrument, with a proper, regular calibration scheme in place (periodically, with at least two reference cylinders, preferably more) for a longer time, with a Picarro, or with cylinder air input, and then show the uncertainty (in ppm / ppb), stability and linearity (!) of the system.*

Thank you very much for your time and careful consideration of our manuscript and all the critical comments, which will vastly improve the final version if the manuscript in case we get the chance the further work on this submission. We are very sorry that the title of the manuscript was misleading – definitely the "atmospheric level measurements" was not the appropriate statement. We initially aimed to test if our system is suitable for this aim, and had to show in the manuscript that the precision for this is too low. But unfortunately we have not reviewed the manuscript title at the end. We are mostly using the system not for monitoring slight atmospheric variations but for determination of environmental GHGs fluxes, like soil chamber measurements, laboratory incubation studies, fermentation experiments.

We definitely agree with the reviewer's opinion that we have incorrectly titled this work and made a wrong emphasis on the atmospheric level measurements. This could be one reason that the manuscript is perceived as very misleading and suggesting the new GC system for long-term GHG's measurements at ambient levels. But definitely our GC system is not dedicated specially for ambient air, but could be useful for determination of possible sources of atmospheric GHGs.

First of all, the levels of precision we have achieved are certainly not even close to those presented in the work of van der Laan's et al (2009). Sorry, we have not found this publication before, in case we would get the chance to prepare the corrected version of our manuscript, this citation will be added and the amazingly high precision of this system will be emphasized. Our method was not designed according to official, WMO recommendations – it is rather the method for fast and cheap detection of GHG's levels at wide range of concentrations. It can be used in agricultural and soil science as well for checking the samples of unknown origin, to determine the presence and concentrations of GHG in range from 0.4 ppm to 0.4% using BID. Using TCD it can detect higher concentrations of GHG's up to 100%. This is the main advantage of proposed system – it is a very simple and relatively cheap application with high flexibility (wide concentration ranges can be measured).

The method which we proposed can have very wide application. Moreover, since the idea is quite new and the system is developed in the past year we would like to share this idea with scientific community at the current state of development, because we know that there is an interest in some research groups to work on similar solutions. If we are allowed to prepare the corrected version of this manuscript we will emphasise that the precision is not sufficient for ambient measurements, method is not perfect and needs further developments, and will give some recommendation of possible enhancements. We believe that even an information on the low precision of this method can be useful for the community – if anybody would have a similar idea, in our manuscript he/she can find a base on how to make such measurements possible and which precision is attainable. It may help to decide in advance if such performance is adequate for his/her aims or not. Such scientific papers are also important, because we bought the system actually as able to reliably measure GHGs, but the exact precision was not known. Since we have tested this for the first time, we believe that it is worth publishing , so that the researchers could check in the literature if the promises of the instrument suppliers are true.

We have to kindly disagree with the statement that our GC system could not be applied in any science or industry with success. We tested this system for soil gases emissions as well as gases from incubations experiments (lignite matter biodegradation experiments or denitrification experiments), samples collected from different industrial sources – gas emissions form wastewater of yeast factory, air samples form Upper Silesia in Poland (coal mining region). Every time before sequence of samples measurement we used calibrated gases directly connected to autosampler (1. Certified reference standard gas from Messer, 2. Compressed air). In the corrected manuscript we will emphasise these applications and provide practical examples. As a response for the questions of reviewer, we attached below the example of calibration lines with information regarding $R^2$ and linearity of calibration line, used for test in this study. This will be also added in the corrected version of the manuscript. We believe that this system has significant potential in the future not as the world's most precise instrumentation for ambient measurements but as simple device for detection of GHG's gases in large range of possible concentrations, hence dedicated for flexible and distinct applications.

**==== Shimadzu LabSolutions Calibration Curve ====**

ID#                          : 1
Name                         : CH4
Quantitative Method          : External Standard
Function                     : f(x)=12311.1*x-4753.88
   Rr1=1.000000  Rr2=1.000000  RSS=2.448455e-021
   MeanRF: 1.081909e+004  RFSD: 1.094662e+003  RFRSD: 10.117873
FitType                      : Linear
ZeroThrough                  : Not Through
Weighted Regression          : None
Offset Correction            : Off
Detector Name                : BID1

[Figure]

| #  | Conc.(Ratio) | MeanArea | Area   |
|----|--------------|----------|--------|
| 1  | 1.895        | 18576    | 19012  |
|    |              |          | 18745  |
|    |              |          | 18159  |
|    |              |          | 18387  |
| 2  | 10           | 118357   | 118483 |
|    |              |          | 118292 |
|    |              |          | 118202 |
|    |              |          | 118452 |

ID#                          : 2
Name                         : CO2
Quantitative Method          : External Standard
Function                     : f(x)=10996.0*x+83666.7
   Rr1=1.000000  Rr2=1.000000  RSS=1.734723e-017
   MeanRF: 1.113963e+004  RFSD: 2.371616e+002  RFRSD: 2.128990
FitType                      : Linear
ZeroThrough                  : Not Through
Weighted Regression          : None
Offset Correction            : Off
Detector Name                : BID1

[Figure]

| #  | Conc.(Ratio) | MeanArea | Area     |
|----|--------------|----------|----------|
| 1  | 411          | 4603028  | 4573051  |
|    |              |          | 4593812  |
|    |              |          | 4643402  |
|    |              |          | 4601846  |
| 2  | 1000         | 11079679 | 10722396 |
|    |              |          | 10855456 |
|    |              |          | 11335187 |
|    |              |          | 11405676 |

ID#                          : 3
Name                         : N2O
Quantitative Method          : External Standard
Function                     : f(x)=9955.50*x-443.654
   Rr1=1.000000   Rr2=1.000000   RSS=1.861156e-024
   MeanRF: 9.296704e+003   RFSD: 7.011233e+002   RFRSD: 7.541633
FitType                      : Linear
ZeroThrough                  : Not Through
Weighted Regression          : None
Offset Correction            : Off
Detector Name                : BID1

[Figure]

| # | Conc.(Ratio) | MeanArea | Area |
|---|---|---|---|
| 1 | 0.339 | 2931 | 2962 |
|   |       |      | 2934 |
|   |       |      | 2966 |
|   |       |      | 2864 |
| 2 | 50 | 497331 | 495410 |
|   |    |        | 499285 |
|   |    |        | 496310 |
|   |    |        | 498320 |

C:\LabSolutions\Data\Michal\20240307\wzorce testy split 3-4-5\Carboxen_SRI_split4_2_pkt_kal_240304.gcm

---

## Author Response (AR1)

**FACULTY OF EARTH SCIENCES AND ENVIRONMENTAL MANAGEMENT**

INSTITUTE OF GEOLOGICAL SCIENCES
pl. M. Borna 9
50-204 Wrocław | Poland

Tel. +48 71 321 10 76
Fax +48 375 93 71

*sekretariat.ing@uwr.edu.pl | www.ing.uni.wroc.pl*

Wrocław, 13.12.2024

Dear Dr. Yoshiteru Iinuma,

Editor,

Atmospheric Measurement Techniques,

On behalf of my co-authors, I am pleased to submit revised version of the article entitled: "Simultaneous measurement of greenhouse gases ($CH_4$, $CO_2$ and $N_2O$) using a simplified gas chromatography system" by: Michał Bucha, Dominika Lewicka-Szczebak, Piotr Wójtowicz

Thank you very much for giving us the opportunity to provide the modified version of our manuscript. We did our best to incorporate all the reviewers comments and suggestions into the final manuscript. The critical reviews helped us to vastly improve our work and clearly indicate the possible applications of our new presented set-up. We are quite convinced that our work will be interested to other scientists working with GC measurements, since this presents new idea for alternative GC system, which has been not described in the literature before.

Please find our point-by-point response to the reviews including a list of all relevant changes made in the manuscript. No part of this manuscript has been published previously and all authors are aware of and accept responsibility for the manuscript.

Best regards,

Dr. Michał Bucha

Institute of Geological Sciences, University of Wrocław, Max Born Sq. 9, 50-204 Wrocław, Poland

E-mail: michal.bucha@uwr.edu.pl

Reply to Reviewer #1

Thank you very much for your positive evaluation on our manuscript and the critical comments which helped us to prepare the improved version of our work.

Please find below our responses and clarifications (black font) and the changes that have been made in the manuscript (blue font).

Page 3, line 89

*What SRI stand for? please write first*

This is the name of the producer, we clarified exact name:

AS-210 Greenhouse Gas Autosampler (SRI Instruments Europe GmbH, Bad Honnef, Germany)

Page 3, line 90

*Molecular sieve*

Yes, the own name of the column was used. This sentence is now corrected with precise information :

The GC separation columns used in this study were performed with a porous layer open tubular column (Carboxen 1010 PLOT) and a molecular sieve column (RT-MSieve 5A), which assured the full separation of the analysed gases.

Page 3, line 95

*The schematic configuration of GC system for simultaneously measurement of CO2, CH4, and N2O should be provided to get better understanding of the this measurement*

Thank you for this suggestion, this will be definitely helpful for the readers. The following scheme is added to Section 2.1, as Figure 1 as well Figure 2 which shows the details regarding flow parameters.

[Figure]

**Figure 1: Configuration of GC system for measurement of CH$_4$, CO$_2$ and N$_2$O**

[Figure]

**Figure 2: Detailed flow parameters in GC system configuration**

Page 3, line 115

*The statement of high sensitivity of BID and the usage of discharge flow, is there any references? (please cited in this statement)*

This is information from Shimadzu Instruction Manual. The respective citation is added: Barrier Discharge Ionization Detector for GC-2010 Plus BID-2010 Plus Instruction Manual. Shimadzu Corporation 2013

Page 4,line 99

*Liquid N2 (LN2)*

The sentence is corrected:

The gas chromatograph oven was equipped with an additional cryogenic option (CRG) where liquid nitrogen (LN$_2$) was used as a cooling agent, which allowed for separation at below-ambient temperatures.

Page 4, line 107 and 108

*Column has length 30 m, 0.32 mm inner diameter (i.d). Please provide also the information of thickness of stationery phase?*

This missing information is now added and corrected as below:

Additionally, using a Tt-joint, the injection sample was then divided between two porous layer open tubular capillary columns filled molecular sieve 5A (RT-Msieve 5A 30 m x 0.32 mm x 30 µm; Restek, USA, Cat. No. #19722) and fused silica (Carboxen 1010 PLOT 30 m x 0.53 mm x 30 µm; Supelco, USA, Cat. No. #25467).

Page 4, line 113

*Porous-layer open-tubular?*

Yes, sentence and phrase are corrected to:

The presence of the traps protected the detectors from particles dislodging from the porous layer open tubular (PLOT) capillary column, which can cause spikes.

Page 4, line 124

*Please express with the consistence significant figures of the mL value (two digit or one digit behind coma)*

The values of flow parameters are corrected to one digit behind coma.

Page 4, line 128

*the standard gas mixtures used in this study, is the certified standard gas mixtures or in-house standard gas mixtures developed by your institute?*

Description regarding standards is improved:

Standard gas mixtures used for testing and final determination of the measurements precision were atmospheric air from Wrocław (Poland) (analyses of $N_2$, $O_2$, $CH_4$, $CO_2$ and $N_2O$ at ambient atmospheric concentrations) and a special gas mixture from Messer ($CH_4$ 10 ppm, $CO_2$, 1000 ppm, $N_2O$ 50 ppm, diluted in pure $N_2$). The in-house standard of compressed air from Wrocław (Poland), which contained natural moisture (vapour), was stored in the 10 litres gas cylinder. The second standard was ordered in Messer Polska Sp. z o.o. and is the commercial product prepared in Switzerland according to the norm ISO6141:2015. This standard was prepared in pure $N_2$, without moisture, in volume of 8 litres and contains F10 filter, which protects outer valve from the possible vapour or solid particles.

Page 4, line 129:

Concentrations are the same as showed in the second columns in the Tables 1 and 2. The link to the values in respective tables will be added (Exact values in Table 1 and 3).

| Gas | Concentration [% or ppm] |
|---|---|
| $N_2$ | 78.084 % |
| $O_2$ | 20.946 % |
| $CH_4$ | 1.895 ppm |
| $CO_2$ | 411 ppm |
| $N_2O$ | 0.339 ppm |

Page 5, line 149

*The peak label (name) for identification of gas component cannot be seen clearly (too small) both in the picture and inset of picture*

Page 6, line 163

*The peak label (name) for identification of gas component cannot be seen clearly (too small) both in the picture and inset of picture*

The figures have been modified for better clarity and larger fonts were applied

[Figure]

**Figure 3: A - Chromatograms of ambient air gases separated using the SH-Q-BOND column and detected using BID; B – Zoomed chromatogram from Figure 3A**

[Figure]

**Figure 4: Chromatogram of special gas mixture separated using the Carboxen 1010 PLOT column and detected using BID**

[Figure]

**Figure 5: Chromatogram of ambient air separated using the RT-MSieve 5A column and detected by TCD**

Page 6, line 164

*Is the compressed air standard mixtures contained the moisture (dry or wet compressed air standard mixtures).*

*How the compressed air standard mixtures prepared should be explained in Materials and method section?*

We clarified this question in section 2.3 as below:

The in-house standard of compressed air from Wrocław (Poland), which contained natural moisture (vapour), was stored in the 10 litres gas cylinder. It was prepared by oil-free compressor for diving cylinders.

Page 6, line 178

*In this section, can O₂ and N₂ be separated well by the GC system and detected by TCD? because in section 3.1, the O₂ and N₂ peaks overlapped when analyzed with GC BID as seen in the chromatogram in Figure 2.*

The separation is showed on newly added figure 5.

*Are the conditions and setup of the GC system in sections 3.1 and 3.2 different?*

Conditions and setup of testing with SH-Q-BOND were exactly the same (injector, TCD, BID configuration, presence of MSieve 5A, the splitting), the only difference was oven temperature programme which was as below:

| Rate | Temperature | Hold Time |
|---|---|---|
| - | 35.0 | 3.50 |
| 10.00 | 130.0 | 0.50 |

The following scheme was added to supplementary materials.

The notification regarding this scheme is added to the text in the section 3.1

[Figure]

*The separation chromatogram for analysis in the GC system in section 3.2 can be displayed*

It is showed in the Figure 5.

Page 7, line 194

*For the better understanding, the data can be added with split ratio of 1,2 at 2 mL sample loops*

We decided not to do it, because the amount of water vapour which can be transferred through the column after injection of 2 ml of ambient air sample would be destructive for the column's filling phase (will shorten its proper functioning). Material used for Carboxen columns is known as very sensitive for water vapour – in the past its filling was used as water adsorber – please see:

Fastyn P., et al. 2003: Adsorption of water vapour from humid air in carbon molecular sieves: Carbosieve S-III and Carboxens 569, 1000 and 1001 - Analyst (RSC Publishing)

Page 7, line 189

*The detection limit of simultaneous analysis $N_2O$, $CH_4$, $CO_2$ using GC BID can be provided further to get the information of the characteristics of this GC system*

In case of $CH_4$ and $N_2O$ we are working at very low levels for BID, almost at the limit of detectability. For $CH_4$ it is 1.8 ppm and for $N_2O$ 339 ppb. The precision error of $N_2O$ at such low concentration is 5%. We are aware that this precision is not really satisfactory for measuring slight variations of ambient levels, however can be very well applied to determine environmental fluxes. We are still working on improvements,  and we will test some modifications few more modifications , e.g. installing additional valve to allow for more gas measurements combinations and temperature controlling of sample loop and injection valves to better control the water level. With this manuscript we intended to publish the first idea of the system with Carboxen column and BID detector already applicable for many environmental studies. Hopefully in near future we can report better precision.

Page 7, line 194

*Why the special gas mixture used in this analysis do not contain the same matrices as compressed air or ambient air (it means the $CH_4$, $CO_2$, and $N_2O$ is in air matrices)*

This results from the process of preparing the calibration mixture by the manufacturer. To obtain the appropriate accuracy, the manufacturer uses pure gas that does not contain the analyte, which is to be added in high dilution.

Our aim was to use the GC system also for samples from  laboratory incubation studies with partially anoxic conditions. For such experiments it is important to determine the low oxygen levels, therefore we needed a gas standards with no and low O2 concentration. To reduce the amount of necessary standards tanks we simultaneously varied O2/N2 levels and levels of GHGs possibly widest range of concentrations of all gases with lowest amount of tanks.

Page 10, line 238

*How about the comparison of precision from this GC system and CRDS Picarro in $CO_2$ and $CH_4$ measurement (standard deviation, %relative standard deviation, error)?*

For these comparison measurements we could only determine % error, because we measured atmospheric ambient air and it shows natural diurnal variations, we compared the results point by point and not the means and standard deviations. This could be only done this way because CRDS Picarro is dedicated mostly to ambient air samples, and it is not possible to insert special gas samples.

Page 10, line 248

*Regarding the upper detection limit, How much the upper detection limit of this simultaneous measurement of $CH_4$, $CO_2$, $N_2O$ by the GC system in this study?*

We can state at this moment that concentrations which can be measured using our GC system are:

$CH_4$ 1.8 ppm to 4000 ppm (using BID), and 0.2% to 100% (using TCD)

$CO_2$ 411 ppm to 4000 ppm (0.4%) – according to Shimadzu specifications of BID detector

$N_2O$ 0.330 ppm (339 ppb) to 4000 ppm – according to Shimadzu specifications of BID detector

Additional information:

$O_2$ – 0.2 to 100% (using TCD)

$N_2$ – 0.2 to 100% (using TCD)

The information regarding the detection limits is added to the conclusions.

**Response to Reviewer #2**

*General comments*

*This paper reports a method for analyzing three greenhouse gases simultaneously using a relatively simple GC system. The strength of this paper is that the developed method enables us to separate and quantify CH4, CO2, and N2O in air samples on a single column with a single detector. This technique would reduce sample size, time, and resources for the gas analyses. A shortcoming of this paper is that the precision or repeatability of the method is insufficient for atmospheric monitoring at background concentration level. In this regard, I think the title is misleading and should be revised to mean that the method is most suitable for source gases such as soil emission. Another concern is that the results of experiments for optimizing the GC setting are mainly discussed in the context of CV without further consideration of sensitivity of the detector. I believe combination of split ratio and sample size affect the amount/concentration of the target species delivered to the detector. For example, split ratio of 1 with 1-mL sample loop should give peak area that is equal to the area obtained at split ratio of 3 with 2-mL loop. In Table 1, I see results of CO2 and N2 are consistent with this idea, but it is not the case for other gases.*

*In summary, I recommend the publication of this paper after the authors address the issues above and specific points below.*

Thank you very much for your positive evaluation on our manuscript and the critical comments which helped us to prepare the improved version of our work. Yes, definitely we proposed inappropriate title for this manuscript, this will be changed to: **"Simultaneous measurement of greenhouse gases (CH$_4$, CO$_2$ and N$_2$O) using a simplified gas chromatography system".** Our initial idea was to test this system for ambient air measurements and although so far the precision is not sufficient for precisely measure the small atmospheric variations, it can be very well applied for measurements of atmospheric fluxes. We missed to review our title before the final submission.

The answer for questions regarding the sensitivity of the BID detector is based on the analysing S/N (signal/noise) ratio for the selected peak. Generally, S/N above 3 allows for identification of the peak, whereas S/N above 10 allows for quantitative determination of concentration. For 339 ppb N$_2$O analysis at split ratio 5 we achieved the ratio S/N usually between 12-15. The S/N ratio for 50 ppm N$_2$O standard was usually above 250. The method how we calculated SD and CV [%] is described in the detailed answers to Reviewer's comments. We presented the equations and method of obtaining the CV [%] for this data. The very high peak area of O$_2$ (Table 1, sample loop 2 ml, split 1) was incorrect (mistake during preparation of the final table) and the corrected value 4513907 will be inserted in the table, which is similar to value 4317080 obtained with split 1 and sample loop 1.

Generally, transferring of the sample from the columns should be very fast to avoid the flattening of the peaks. The sample is transported from the sample loop to the column at total flow speed. Total flow is mainly dependent on the split value: total flow = column flow + split ratio x column flow + purge flow (3ml/min). Therefore, the larger the split value, the faster the sample reaches the column. The narrower band in which the sample hits the column makes the peaks narrower and higher than with lower split values, even though less sample hits the column. However, this is one of the rules that must be checked experimentally each time to find appropriate values for the experiment being conducted.

In this method we also consider the amount of water vapour, which is transferred to the column. It is important to find the ideal compromise between the appropriate amount of gas supplied to the column and obtaining a strong signal, especially for the lowest concentrations using BID. Therefore, we assume that the peak areas are not consistent with idea of the Reviewer mainly due to the difference in the speed of transferring through the columns. The additional calculation is showed below to compare all uncertainties. The difference between obtained peak areas is showed in the table as column C, and its absolute value in column D. Then in column E we showed that the value A (difference of the peak area expressed in %) is much higher for smaller peaks ($CH_4$ and $N_2O$, 23.30 and 28.66%, respectively). For larger peaks of $CO_2$, $O_2$ and $N_2$ the difference E is equalled around 5%. Therefore, it is very important to monitor parameter of S/N ratio when analysing low concentration samples.

| Gas | A
Peak area at split 1 sample loop 1mL | B
Peak area at split 3 sample loop 2mL | C
C=A-B | D
Absolute value of C | E [%]
E=(A*100)/D |
|---|---|---|---|---|---|
| $CH_4$ | 33294 | 25535 | 7759 | 7759 | 23.30 |
| $CO_2$ | 10900323 | 10274295 | 626028 | 626028 | 5.74 |
| $N_2O$ | 7801 | 5565 | 2236 | 2236 | 28.66 |
| $O_2$ | 4317080 | 4513507 | -196427 | 196427 | 4.55 |
| $N_2$ | 16522678 | 17338000 | -815322 | 815322 | 4.93 |

Please find below our responses to the specific points and clarifications (black font) and the changes that have been made in the manuscript (blue font).

*Specific comments*

*L40–41. I think "the most modern techniques and devices" should be adopted only if they provide results with precision and accuracy that are sufficient for the purpose.*

We agree with this comment and we are aware it is not suitable for measuring minimal changes in greenhouse gas concentrations in ambient air over time (and such measurements are required for monitoring climate change). Therefore we decided to change the manuscript title which is now:

"Simultaneous measurement of greenhouse gases ($CH_4$, $CO_2$ and $N_2O$) at atmospheric levels using a simplified gas chromatography system"

We also improved the description of our GC configuration, showed possible applications and underline its strengths and weakness.

*L46. I think a chapter from an e-book is referred here. Correct the citation information in the References section.*

You have right. This is a chapter from the e-book. Citation is corrected in the reference list.

Zaman, M., Kleineidam, K., Bakken, L., Berendt, J., Bracken, C., Butterbach-Bahl, K., Cai, Z., Chang, S.X., Clough, T., Dawar, K., Ding, W.X., Dörsch, P., dos Reis Martins, M., Eckhardt, C., Fiedler, S., Frosch, T., Goopy, J., Görres, C.-M., Gupta, A., Henjes, S., Hofmann, M.E.G., Horn, M.A., Jahangir, M.M.R., Jansen-Willems, A., Lenhart, K., Heng, L., Lewicka-Szczebak, D., Lucic, G., Merbold, L., Mohn, J., Molstad, L., Moser, G., Murphy, P., Sanz-Cobena, A., Šimek, M., Urquiaga, S., Well, R., Wrage-Mönnig, N., Zaman, S., Zhang, J., Müller. C., 2021. Methodology for Measuring Greenhouse Gas Emissions from Agricultural Soils using Non-Isotope techniques. 11-209. In: Zaman M, Kleineidam K, Bakken L, Berendt J, Bracken C, Butterbach-Bahl K, Cai Z, Chang SX, Clough T, Dawar K, Ding WX, Dörsch P, dos Reis Martins M, Eckhardt C, Fiedler S, Frosch T, Goopy J, Görres C-M, Gupta A, Henjes S, Hofmann MEG, Horn MA, Jahangir MMR, Jansen-Willems A, Lenhart K, Heng L, Lewicka-Szczebak D, Lucic G, Merbold L, Mohn J, Molstad L, Moser G, Murphy P, Sanz-Cobena A, Šimek M, Urquiaga S, Well R, Wrage-Mönnig N, Zaman S, Zhang J, Müller C (2021) Measuring Emission of Agricultural Greenhouse Gases and Developing Mitigation Options Using Nuclear and Related Techniques Springer ISBN 978-3-030-55395-1, https://doi.org/10.1007/978-3-030- 55396-8.

*L53–55. I think mass spectrometer is one of the detectors used in gas chromatography. I cannot understand why the mass spectrometry is specifically noted.*

You have right, we deleted the part of the sentence "gas chromatography and/or gas chromatography coupled with mass spectrometry (GC-MS)". Corrected sentence is:

Thus, the most reliable methods for GHGs measurements in a very wide range of concentrations are chromatographic methods (Ekeberg et al., 2004).

*L56–63. This paragraph is difficult to read because there is a lot of duplication. It seems that the authors try to list several types of "systems", but they just mention detectors and GC columns.*

You have right. We corrected and improved this paragraph to avoid duplication.

Gas chromatography with automated sampling injections is a very common, flexible and user friendly technique. The most common GHG measurement systems have been developed with: a thermal conductivity detector (TCD) (measurement of $CH_4$ and $CO_2$), flame ionisation detector - FID (measurement of $CH_4$ and $CO_2$ using a methaniser), electron capture detector - ECD (for $N_2O$ measurement) (Hedley et al., 2006; Loftfield et al., 1997; Wang and Wang, 2003).

*L64. What does "natural wear" mean? Because the half-life of the radiation source Ni-63 is 100 years, I wonder other factors are meant.*

The ECD detector cell is a consumable part. The cells should be replaced every 2-5 years. Moreover, the ECD is very sensitive to oxygen, which oxidizes the nickel foil. Using poor quality nitrogen, e.g. 5.0, which contains trace amounts of oxygen, is enough to shorten the cell's life. Before the ECD, oxygen traps are used that must be replaced regularly. In addition, the ECD gets dirty with the stationary phase from the column. In general, the thicker the film inside the chromatography column, the faster the cell wears out. The ECD also gets dirty simply with the measured analytes. The ECD cell can no longer be disassembled and cleaned. Theoretically, it can be sent to service for cleaning, but the price is so prohibitive that it is not worth doing (the price of cleaning ~ the price of buying a new cell). The ECD can only be annealed. On the other hand, the BID is a maintenance-free detector, by definition it does not get dirty due to the dielectric barrier.

*L78. It is not clear what "and/or" means. In a certain case, both the two-column system and the single- column system are required?*

You have right, this I misleading phrase. The corrected sentence is:

Separation of $CH_4$, $CO_2$, and $N_2O$ from one sample can be done using, for example, a system of two columns with 10-port valves (Scion Instruments, 2023) or a single column e.g. Micropacked ST Shin Carbon or RT Q-Bond column (Shimbo and Uchiyama, 2022).

*L81–83. This sentence is difficult to understand. For example, what is compared using "as well as"?*

The corrected sentence is:

This set-up using single column and single BID detector is commonly used for determination of CH4 and CO2 at very low atmospheric concentrations (Gruca-Rokosz et al., 2020).

*L95. I recommend to add a schematic figure showing the GC system.*

The new, two figures are added to the text – scheme of GC configuration and details parameters of the flow. Moreover, the scheme of the GC configuration with column SH-Q-BOND is showed in the Appendix and mentioned in the section 3.1.

[Figure]

**Figure 1: Configuration of GC system for measurement of CH₄, CO₂ and N₂O**

[Figure]

**Figure 2: Detailed flow parameters in GC system configuration**

**Appendix 1**

The scheme of GC configuration for testing with columns SH-Q-BOND and RT-Msieve 5A

[Figure]

*L101–102. "warmed" at what temperature?*

It was 110°C. This information is added to the text and is showed on scheme of the GC system.

*L102–103. This means gas sample with high moisture (e.g., soil gas) might cause problems. Is the system designed for already dried samples or does it withstand moist samples?*

We tested Messer standard which doesn't contain water vapour as well environmental samples with natural, high moisture (collected from the soil after rainfall etc.). Generally for samples with

significantly higher concentrations of $CH_4$, $CO_2$ and $N_2O$ presence of moisture does not significantly affect the measurement. However, for samples with ambient concentrations of $N_2O$ and with high moisture the baseline (and noise) should be checked to be sure that the signal of $N_2O$ is enough to measure the concentrations. Shimadzu recommends that S/N (signal/noise) ratio should be above 10 to be sure that determination is correct. For 339 ppb $N_2O$ analysis at split ratio 5 we achieved the ratio S/N usually between 12-15. The S/N ratio for 50 ppm $N_2O$ standard was usually above 250.

The analysis of gas samples with natural moisture will lead to faster decay of the column filling (film) and accumulation of the solid parts in the particle trap, as a result, to a rise of the baseline, which has the greatest impact on the determination of samples with extremely low concentrations. For prevention we are actually testing different moisture traps (connected to the capillary between GC and autosampler). This is to minimize the presence of water vapour inside the system (such as the sample loop).

*L107–110. I guess the length and inner diameter of the columns are described here, but the splitting ratio of 1:5 cannot be achieved with the dimensions shown here. Add other parameters such as thickness of the inner coating of the columns. Also, combination of column and detector should be clearly described or shown using a figure. Is the TCD connected to the molecular sieve column?*

We will modify the description in the text as well showed basic information on GC scheme. The corrected text will be as below:

Additionally, using a T-joint, the injection sample was then divided between two porous layer open tubular capillary columns filled molecular sieve 5A (RT-Msieve 5A 30 m x 0.32 mm x 30 µm; Restek, USA, Cat. No. #19722) and fused silica (Carboxen 1010 PLOT 30 m x 0.53 mm x 30 µm; Supelco, USA, Cat. No. #25467). The dimensions of the columns were selected to achieve a splitting ratio of 1:5, directing most of the sample to the Carboxen 1010 PLOT and BID. Corresponding calculations were performed in Shimadzu AFT (Advanced Flow Technology) software (Fig. 2).

*L111. Quantitative information should be given instead of "extremely low baseline noise".*

We corrected the sentence and added information regarding S/N ratio (signal/noise ratio):

Extremely low baseline noise (signal/noise ratio (S/N) always above 10) was achieved by a combination of two factors: a high purity carrier gas helium grade 5.0 connected to the Valco Helium Purifier HP2 (VICI Valco Instruments Co. Inc.) and particle traps (2.5 m x OD 0.32 mm) mounted on the columns' outlets.

*L115–117. It is not clear at which position the discharge gas is added to the flow system.*

Discharge gas is connected and passed from the top, it is used to create plasma, below is a technical scheme from Shimadzu manual instruction:

[Figure]

Please see also link to the Schimadzu website BID | Research & Development | SHIMADZU CORPORATION with details regarding BID.

*L117. I cannot understand what this sentence means.*

On the TCD channel satisfactory sensitivity was achieved with standard settings - current 80 mA value and make-up gas 8 mL/min. Generally, in TCD, sensitivity can be adjusted by changing the type of carrier gas, make-up and current.

We added citation to the description of the method (information from the instruction manual of GC-2010 Nexis):

Barrier Discharge Ionization Detector for GC-2010 Plus BID-2010 Plus Instruction Manual. Shimadzu Corporation 2013)

*L121–122. Do the authors mean the final temperature of 200C is kept for 1 min? Revise the sentence.*

Yes. The final temperature is called "hold time". The temperature program is also now showed on the figure with GC Scheme.

*L123–124. As described in the previous section, the flow after sample injection was divided into two columns. Are these flow parameters common to them?*

It is necessary to add 2.5m particle traps with a diameter of 0.32mm to the columns. For a column with a diameter of 0.32mm, we added 2.5m which gives 32.5m. The column of 0.53mm should be theoretically extended by 19m, therefore 2.5m x 0.32mm gives the same resistance as 19m x 0.53mm, therefore the second column has a entered length of 49m. The details are showed in the new Figure 2 (previously showed in the response).

*L140. Decrease from what temperature?*

What we mean here is a quick stabilization of the oven temperature. Temperature stabilization. 35° is the lowest temperature that can be practically achieved in our laboratory without using liquid nitrogen for cooling. The temperature reduction that we mention here concerned the standard temperature of 40°C (at which we most often tested the SH-Q-Bond column).

The oven programme and configuration of SH-Q-BOND is showed in the new Appendix 1 (previously showed in the response).

*Figures 1 and 2. Labels on the x and y axis are difficult to read.*

The Figures are corrected with larger size fonts.

*L158. Does "vapour" mean water vapor? Please specify. Also, do the authors mean that the retention time of H2O peak shown in Figure 2 changes depending on the amount of water in the sample?*

Yes, vapour means water vapour (natural moisture). The $H_2O$ peak appeared usually when the oven temperature reached 115°C – it is not perfectly visible because the oven temperature was rising (and thus the baseline level too).

To be more precise, we have corrected in the whole manuscript 'vapour' to 'water vapour'.

*L165. It is not clear to me for what purpose the authors made experiments with different combinations of split ratio and sample size. I think the amount of sample (and water vapor) injected to the column is determined by the two parameters. For example, if 1 mL sample is processed with split ratio of 1, the amount of sample injected to the column is 1×1/(1+1) = 0.5 mL. If 2 mL sample is processed with split ratio of 3, the amount would be 2×1/(1+3) = 0.5 mL, which is the same as the first case.*

We tested different split settings and loop volumes to find a compromise between the amount of sample analyzed and the speed of sample transport to the chromatography column. If we use a 2 mL loop, we can dose a larger amount of sample onto the column, which we want to transport to the column as quickly as possible. If we do it slowly, then the peaks will be broad and low, we will lose sensitivity. Hence, although the mount is identical, the flow through the column will differ between these settings.

*L177 and elsewhere. Since TCD, FID, and ECD are acronyms of "xxx detector", notation like "TCD detector" is awkward.*

Thank you for this comment. We corrected these mistakes.

*Table 1. It seems "SD" does not show the standard deviation of peak area, because dividing this value with "area" gives much smaller CV value. This is also the case for Tables 2 and 3. Please correct.*

You have right. We suppose that these ambiguities resulted from the different way for presentation of the data (SD and CV) and lack of a clear presentation of the used calculation method in the table's legends. In the original manuscript SD showed the standard deviation of the calculated concentration. First, we measured the same standard 3, 4, 5, 10 or 20 times. Then, this raw data of measured peak area was used for calculation of mean peak area typical for concentration at different and known levels (calibration). Concentrations were calculated according to equation:

Concentration [%] = (measured peak area * known concentration of the standard) / (calculated mean peak area from 3, 4, 5, 10 or 20 analyses)

SD = standard deviation of concentrations calculated for 3, 4, 5, 10 or 20 analyses

CV = (SD * 100) / (mean peak area from 3, 4, 5, 10 or 20 analyses)

In the manuscript we corrected all the tables according to Reviewer's suggestion.

*L185. Specify "the main atmospheric gases".*

We mean $O_2$ and $N_2$. We will correct the paragraph because the measurement of these two compounds was performed with TCD – in our system is dedicated to a gases of higher concentrations than 0.2% (2000 ppm). Therefore the comparison of the SD obtained from BID which is much more sensitive in our opinion it is not recommended. New paragraph is corrected:

The gases analysed using the TCD detector, $O_2$ and $N_2$, were characterised by a narrow CV ranging from 0.10 to 0.39 %. The highest CV (0.39 %) was observed for the $N_2$ measurement with the sample loop 2 mL, where the peak area was very large. The results of the measurement (peak area, SD, CV) are presented in Table 1.

*L202. What does "reportable results" mean?*

This was a mistake, most probably made by autocorrection. We should use the phrase "repeatable results" – the achieved precision of $N_2O$ analysis is considered as sufficient for determining environmental gas samples, e.g. from soil experiments and measurements of gas emissions from natural sources such as peat bogs.

*L212–214. If the authors use the system for soil gases, should the sample be dried before analysis?*

No, we recommend to use original sample and eventually trap the water moisture (using sorbents) – but these sorbents needs to be tested in order to check potential influence on the concentrations of other gases (e.g. sorption of $CO_2$ or $N_2O$). The best way is to analyse pure samples, and find the conditions when water moisture does not affect the precision of GC measurements.

*Table 3. Is the area for N2O at split ratio of 3 correct? It is extremely higher than those obtained at split ratio of 4 and 5.*

Thank you for this comment. This is mistake. The values in the table are corrected. The correct value of area= 5552, SD=30.6, CV [%]=9.02.

*L234 split ratio of 4*

It is corrected.

*L235–237. I cannot understand these sentences. Do the authors mean the "GC" results of CO2 obtained after 25/11/23 18:00 (Figure 3) have a systematic error due to a shift in sensitivity of the detector?*

Yes, some shift occurred, eg. due to change in air moisture, this was Friday evening after all people left laboratory. But importantly, with new calibration the both measurements would stay in very good agreement.

One of the reason why Picarro is much more precise could be amount of sample analysed. Picarro analyses at least 20mL of sample for constant measurement, which make a single measurement every few seconds. Our GC system is sampling air in the volume of 1 or 2 mL. By design, such a

measurement will not be as accurate in the long-term run as a device specifically dedicated to analyzing $CO_2$ or $N_2O$ fluctuations in the atmosphere. But simultaneously, our system is not limited to low concentrations of GHG's, which is why it stands out from other special devices.

*L243–245. Although it is not clear what "typical N2O measurements" means, 5% error is NOT satisfactory if one would like to know diurnal/seasonal cycle or secular trend of atmospheric N2O at background level.*

Yes, we agree with this comment. However, for measurement of larger fluxes, like e.g. soil gases such error is accepted by scientific community. Many scientists are working with such precision (as the cited papers), if these measurements are not related to very precise monitoring of ambient N2O concentrations. Also, for higher concentrations our errors are much lower. We have now emphasised this difference in applications in the manuscript.

*L250–251. Since no TCD chromatogram is shown, this statement cannot be verified.*

We added the chromatogram with TCD to the manuscript as below:

[Figure]

**Figure 5: Chromatogram of ambient air separated using the RT-Msieve 5A column and detected by TCD**

*L261. Consider other quantitative or scientific expression for "very good".*

The sentence is corrected:

A single column Carboxen 1010 PLOT can be successfully used for separation of GHG ($CH_4$, $CO_2$, $N_2O$) in time interval enabling measurement of each gas separately without the effect of peak overlapping.

*L268. water vapour?*

Yes, we added clarification. The paragraph is corrected:

Based on performed tests, it is recommended that atmospheric GHGs are analysed using a BID at split ratio 4 or 5 and with a sample loop of 2 mL volume. This would help avoid unnecessary

contamination of the Carboxen column with water vapour, therefore splits 1-3 should not be considered for the measurement of environmental gas samples.

**Response to Reviewer #3**

*I am afraid I have to reject this paper in its present form, simply because to my opinion the instrument in its present state lacks being fit-for-purpose in any application.*

*While the simultaneous detection of the three anthropogenic greenhouse gases using just one detector and only He is an improvement compared to previous GC systems, the precision and accuracy is much too poor for any application, most certainly for atmospheric measurements. Compare for instance to van der Laan et al (2009): A single gas chromatograph for accurate atmospheric mixing ratio measurements of CO2, CH4, N2O, SF6 and CO. Atmospheric Measurement Techniques 2, 549–559 (2009). (not cited in the paper). They use two detectors, FID and EC, but they reach precisions : ±0.04 ppm for CO2, ±0.8 ppb for CH4, ±0.8 ppb for CO, ±0.3 ppb for N2O, and ±0.1 ppt for SF6, which are within the reach of the WMO recommendations for atmospheric monitoring. The present instrument gets uncertainties more than two orders of magnitude worse!*

*If the authors have other applications in mind, soil air or agricultural greenhouse air for example (as several of the papers they cite, and which is hinted at by the values for their reference cylinder with strongly enhanced concentrations compared to ambient), then they should make that very clear in the paper. Furthermore, I think that even for these applications the present quality is not good enough.*

*The authors should compare their instrument, with a proper, regular calibration scheme in place (periodically, with at least two reference cylinders, preferably more) for a longer time, with a Picarro, or with cylinder air input, and then show the uncertainty (in ppm / ppb), stability and linearity (!) of the system.*

Thank you very much for your time and careful consideration of our manuscript and all the critical comments. According to the decision of the Editor of the Atmospheric Measurement Techniques we submitted revised version of the manuscript, which is significantly improved and takes into account all Reviewer's comments. The new title of the manuscript is "Simultaneous measurement of greenhouse gases ($CH_4$, $CO_2$ and $N_2O$) at atmospheric levels using a simplified gas chromatography system".

In the revised version we added scheme of GC configuration and details regarding flow parameters, new chromatograms, revised all the data including calculation of standard deviation, errors, discussed other systems (also proposed by the Reviewer – discussed in the article of van der Laan et. al 2009). In Appendix 1 we showed the GC configuration used for testing of SH-Q-BOND column. We also showed the possible applications of our simplified GC system in the conclusions section. We underline that proposed method is suitable for measurement of wide range GHGs concentrations, not for monitoring the slight changes in ambient air. Therefore, we assume that GC system can be shared to scientific community and help in development of alternative methods for GHGs measurement. We believe that this system has significant potential in the future not as the world's most precise instrumentation for ambient measurements but as simple device for detection of GHG's gases in large range of possible concentrations, hence dedicated for flexible and distinct applications.

We have also tested the system linearity, which was very good in range of the tested standards. This information was added in the manuscript.

In this manuscript we present the new idea of the system construction and the truly attainable precisions, which are indeed not so good for low N2O levels. For higher N2O concentrations and for CO2 and CH4 we get really satisfactory results in the whole measurement range. We believe, although this method do not provide the worlds-best precision, it is still worth publishing, because it presents new idea of the GC system set-up and maybe some scientists would decide this can be applicable for their needs (e.g. for CO2 and CH4 only, or for higher N2O fluxes). It can be also further developed or refined, maybe in future better results can be obtained. Moreover, this is a documentation of how this set-up works – if anybody would like to construct similar system, with our work he can obtain the initial information how this can be done and which precision has been obtained.